# Seismic Data Interpretation and Petrophysical Analysis of Kabirwala Area Tola (01) Well, Central Indus Basin, Pakistan

**Naveed Ahmad** [1], **Sikandar Khan** [2,*] **and Abdullatif Al-Shuhail** [3]

1 Department of Marine Geophysics, Ocean University of China, Qingdao 266100, China; naveedahmadgeophy@gmail.com
2 Department of Mechanical Engineering, King Fahd University of Petroleum and Minerals, Dhahran 31261, Saudi Arabia
3 Department of Geosciences, King Fahd University of Petroleum and Minerals, Dhahran 31261, Saudi Arabia; ashuhail@kfupm.edu.sa
* Correspondence: sikandarkhan@kfupm.edu.sa

**Abstract:** Well logging is a significant procedure that assists geophysicists and geologists with making predictions regarding boreholes and efficiently utilizing and optimizing the drilling process. The current study area is positioned in the Punjab Territory of Pakistan, and the geographic coordinates are 30020′10 N and 70043′30 E. The objective of the current research work was to interpret the subsurface structure and reservoir characteristics of the Kabirwala area Tola (01) well, which is located in the Punjab platform, Central Indus Basin, utilizing 2D seismic and well log data. Formation evaluation for hydrocarbon potential using the reservoir properties is performed in this study. For the marked zone of interest, the study also focuses on evaluating the average water saturation, average total porosity, average effective porosity, and net pay thickness. The results of the study show a spotted horizon stone with respect to time and depth as follows: Dunghan formation, 0.9 s and 1080.46 m; Cretaceous Samana Suk formation, 0.96 s and 1174.05 m; Datta formation, 1.08 s and 1400 m; and Warcha formation, 1.24 s and 1810 m. Based on the interpretation of well logs, the purpose of petrophysical analysis was to identify hydrocarbon-bearing zones in the study area. Gamma ray, spontaneous potential, resistivity, neutron, and density log data were utilized. The high zone present in the east–west part of the contour maps may be a possible location of hydrocarbon entrapment, which is further confirmed by the presence of the Tola-01 well.

**Keywords:** 2D seismic interpretation; contour mapping; petrophysical analysis; porosity

## 1. Introduction

Well logging is a significant tool that assists geophysicists and geologists with making predictions regarding hydrocarbon reservoirs and provides important information such as rock composition, characteristics, and integrity as well as the presence of fluids and hydrocarbon. The collected data from well logs can be utilized to perform feasibility studies relevant to the physical properties of the area of interest under geophysical work. The oil and gas industries utilize estimated petrophysical parameters from well log data, which is a crucial part of the exploration and production processes [1,2]. These petrophysical parameters also assist with interpreting the subsurface for detection and estimation of hydrocarbon reservoirs. Furthermore, the oil and gas industries mostly use wireline logging to achieve continuous records regarding formation rock properties [3–5].

Seismic data interpretation is commonly used for hydrocarbon exploration and development studies [6–9]. The interpretation of seismic data requires an understanding of the subsurface formations and wave propagation effects. Seismic data consist of reflections that represent the geological structures as well as the reservoir stratigraphy, fabric, and fluid content. Al-Shuhail et al. [3,5] presented 2D viscoelastic models of the Ghawar and central fields of Saudi Arabia and created corresponding multi-component synthetic seismic

datasets for hydrocarbon seismology exploration. Similarly, petrophysics plays a significant part in the chemical and physical properties of rocks and their internal fluids. Petrophysical data are generally achieved from well logs and can be used for qualitative and quantitative analysis. The evaluation of petrophysical analysis of well logging provides core properties such as lithology, grain size, porosity, water saturation, clay volume, and permeability [10]. Petrophysics plays an important role in seismic interpretation and has experienced a great leap forward in the past 10 years from significant advances in seismic data processing practices, including full waveform inversion, attribute analysis, and amplitude versus offset methods used in the estimation of reservoir properties from pre-stack and post-stack seismic data [7]. The findings of the literature review are summarized in Table 1.

The main objective of the current research is to interpret the subsurface structure and reservoir characterization of Kabirwala area Tola (01) located in Punjab platform, Central Indus Basin, utilizing 2D seismic and well log data. The study area (Tola (01) well) is in the Central Indus Basin (CIB). The Central Indus Basin (CIB) consists of Punjab Platform, Suleiman Depression (Zindapir Inner Fold Zone, Mari Bugti Inner Fold Zone), along with the Suleiman Fold Belt. As shown in the below Figure 1a, the Tola (01) well is situated in the Kabirwala area near district Multan. The Kabirwala extent is positioned in the Punjab Territory, the geographical coordinates are 30020′10″ North and 70043′30″ East. Kabirwala is one of the four tehsils in Khanewal district Multan. It lies in Central Indus Basin Punjab Platform, Pakistan. Five seismic lines—875-KBR-230, 875-KBR-229, 875-KBR-231, 875-KBR221, and 875-KBR-220—and well logs of Tola-01 were used to explain the subsurface structures and demarcation of zones that have fair potential for hydrocarbon accumulation. The main aim is formation evaluation for hydrocarbon potential using the reservoir properties. In addition, time and depth contour maps are generated to delineate lateral extension and closure of the reservoir. Based on well log interpretation, the main aim of petrophysical analysis is to recognize hydrocarbon-bearing zones in the projected study area. The logging data of gamma ray, spontaneous potential, resistivity, neutron, and density were utilized, as shown in Figure 1a. The exploration license of Tola (01) was granted to Oil and Gas Development Company Limited (OGDCL) in June 1992. The acquisition and processing of 2D seismic data took place on Fort Abbas field in 1994. Exploratory wells Tola-01 and Nandpur-01 were drilled in 1994 and 1996, respectively, on the Punjab platform to discover the petroleum potential of Infra Cambrian reservoir rocks. The input data used in the current research study were taken from OGDCL.

**Table 1.** Findings of literature review.

| Reference | Parameters Used | Simulation Method (2D, 3D, 4D) | Petrophysical Analysis Used (Yes/No) | Stratigraphy Interpretation Used (Yes/No) | Objective of Research Study |
|-----------|-----------------|--------------------------------|--------------------------------------|-------------------------------------------|------------------------------|
| [11] | Volume of shale, porosity, hydrocarbon saturation, water saturation | 3D | Yes | Yes | Discovery of hydrocarbon reserve resource accumulations in Pennay field |
| [12] | Effective porosity, clay content, and water saturation | – | Yes | Yes | Proposal of joint estimation of petrophysical properties combining statistical rock physics and Bayesian seismic inversion |

**Table 1.** *Cont.*

| Reference | Parameters Used | Simulation Method (2D, 3D, 4D) | Petrophysical Analysis Used (Yes/No) | Stratigraphy Interpretation Used (Yes/No) | Objective of Research Study |
|---|---|---|---|---|---|
| [13] | Volume | 3D | Yes | Yes | Estimation of brittleness of resource plays in 3D by integrating petrophysics and seismic data analysis |
| [14] | Joint probability function, porosity, thickness | 3D | Yes | Yes | Estimation procedure for extrapolating wireline data from existing wells using geostatistical inversion of post-stack 3D seismic data |
| [15] | Porosity, water saturation and hydrocarbon saturation | 3D | Yes | Yes | Determination of reservoir properties and estimated volume of hydrocarbons within reservoirs |
| [16] | Porosity, net-to-gross, water saturation, hydrocarbon saturation | 3D | Yes | Yes | Reserve evaluation of hydrocarbon-bearing sands |
| [17] | Volume of shale, porosity, water saturation, formation water resistivity, and hydrocarbon saturation | 2D | Yes | Yes | Structural interpretation and hydrocarbon potential of Balkassar oil field |
| [18] | Density, neutron, self-potential, and resistivity | 2D | Yes | Yes | Structural interpretation of Joya Mair oil field using 2D seismic data and petrophysical analysis |
| [19] | Porosity, permeability, moderate net to gross, and low water saturation | 3D | Yes | Yes | Application of 3D static model using 3D seismic and well log data for optimization and development of hydrocarbon potential in KN field of Niger Delta Province |

The current research work is organized as follows: Section 2 describes the tectonics of the proposed area, Section 3 describes the generalized stratigraphy of the area, and Section 4 presents a comprehensive analysis of the seismic interpretation and the well log analysis, Section 5 consists of a detailed petrophysical analysis, Section 6 presents the discussion, and finally, Section 7 presents the conclusions of the current research work.

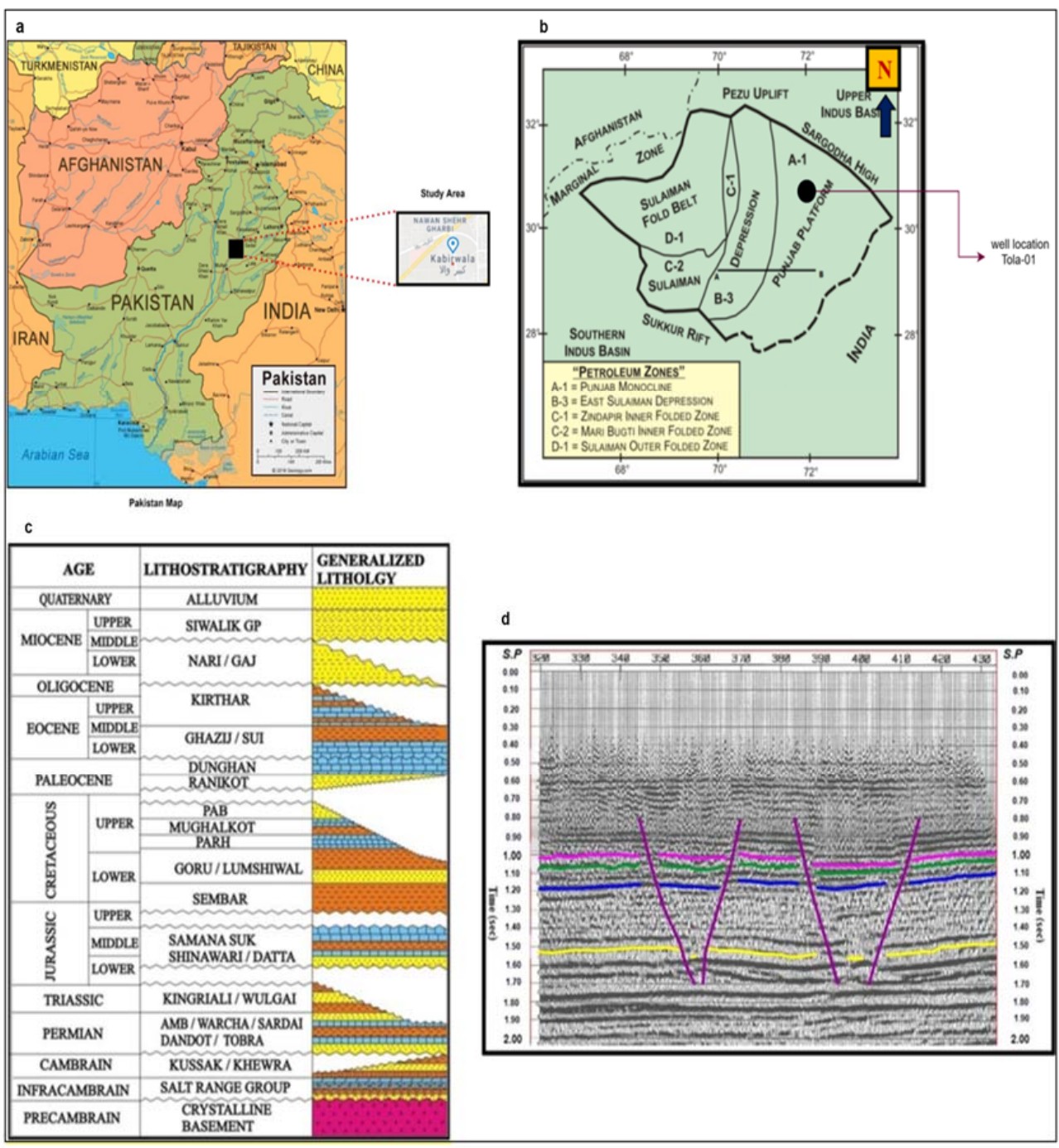

**Figure 1.** (**a**) Location of the study area. (**b**) Tectonic map of Central Indus Basin and subdivisions of petroleum zones (after [20]). (**c**). Generalized stratigraphic chart of the study area [21]. (**d**). Seismic section with marked horizon.

## 2. Tectonics of the Area

There are three subdivisions of the Central Indus Basin (CIB): the Punjab platform, Sulaiman fold belt, and Sulaiman depression. The Punjab platform is located at the east–west of the Sulaiman depression and fold belt, as shown in the structural map in Figure 1b [8]. The Punjab platform is a westward-dipping monocline covered with unconsolidated Quaternary deposits with a maximum thickness of about 500 m [9], and the area is almost all desert and neglected surface outcrops [10]. As shown in Figure 1b, Sargodha high is at the north of the CIB, Indian Shield is at the east, an axial belt is at the west, and finally Sukker Rift is located at the south [10]. The Punjab platform was the least affected during

Himalayan orogeny; therefore, it possesses non-tectonic origin structures [11]. It also has no exposure to bedrocks. The primogenital rock existing in the basin is counted in the Triassic age. Owing to the absence of active tectonic elements in the CIB, it is almost stable [12].

## 3. Generalized Stratigraphy of the Area

Stratigraphy is mainly concerned with the study of rock layers (strata) and layering (stratification). Stratigraphy has two related subfields: lithostratigraphy and biostratigraphy. The generalized stratigraphy of the study area is discussed in the following subsections.

### 3.1. Pliocene/Miocene Stratigraphy

The Pliocene/Miocene stratigraphy of the area includes the following formations:

#### 3.1.1. Nagri Formation

Its lithology is constituted of sandstone with dependent clay as well as conglomerate. Its topmost association is with Dhok Pathan formation, which is transitionary. The age is Late Miocene.

#### 3.1.2. Chinji Formation

Its lithology constitutes of red clay with dependent ash gray or brownish-gray sandstone. It is exclusively limited to the southern bisection of the eastern Suleiman range and is not elaborated in the repose of the Lower Indus Basin. It is substantially superimposed beside the Nagri formation. The age is Late Miocene [21].

### 3.2. Oligocene Stratigraphy

The Oligocene stratigraphy of the area includes the following formation:

#### Nari Formation

The name "Nari series", derived from the Nari River in the Kirthar range, was introduced by Khokhar et al. [22]. The Nari formation consists of sandstone, shale, and subordinate limestone. The lower part of the formation, the Nal member, is predominantly composed of crystalline limestone, which is white to gray, brown or yellow shelly, nodular or rough surfaced, thin to thick bedded, and even massive. Thin stringers of dark shale and of thin bedded and fine-grained brown sandstone are also present.

### 3.3. Eocene Stratigraphy

The Eocene stratigraphy of the area includes the following formations:

#### 3.3.1. Habib Rahi Formation

The Habib Rahi formation was formalized by the Stratigraphic Committee from Habib Rahi limestone of Tainsh et al. [23]. It can be described as a grayish-brown and buff hard limestone that is weathering white. It is fine-grained, thin-bedded, dominantly argillaceous, and in a few places grades into marl.

#### 3.3.2. Pirkoh Formation

The Pirkoh limestone member was introduced by Hemphill and Kidwai [24]. Rehman et al. [25] defined the formation as follows: lithologically, the Pirkoh formation consists of limestone that is light gray to chalky white, buff to brown, fine-grained, mostly thin and regularly bedded, and often argillaceous. The formation commonly contains subordinate beds of soft, shaly limestone, and dark gray calcareous claystone.

### 3.4. Paleocene Stratigraphy

The Paleocene stratigraphy of the area includes the following formations:

### 3.4.1. Ghazij Sui Member

The overall lithology in the early Eocene episode is shale, as well as marl in the Lower Indus Basin, along with adjacent areas of the axial belt. The Ghazij Sui member consists of Kirthar formation. The Kirthar formation is separable into four components in sections of eastern Sulaiman province as Habib Rahi limestone, Sirki, Pir koh, and Drazinda components [26].

### 3.4.2. Dunghan Formation

The Dunghan formation was familiarized by Kazmi [27]. Malkani et al. [28] nominated the order portion to be adjacent to Harnai (latitude 300 08′38″ N, longitude 670 59′33″ E) and renewed the close rank Dunghan formation. It is constituted of limestone, shale, and marl. The limestone is gray to yellowish, fine to intermediate bedded with conglomerates. The shale is gray and khaki as well as calcareous. The marl is brown to gray, fine to intermediate bedded, and fine-grained. Sideward, these formational facies are dissimilar at places with dense limestone precipitates, while at some positions, there is insignificant limestone evidence. The Sui main is the topmost portion of the Dunghan limestone because of its fluctuating manner. It is dense in the Zindapir, Duki, Sanjawi, Harand, and Mughal Kot portions, but it is minute in the Rakhi Gaj and in Mekhtar regions. Petroleum evidence is common in this formation, notably in the Khatan portion [27]. Adjacent to the axial belt, it has disconformity at the base, while the topmost association with the Shaheed Ghat formation is transitionary. It has many mega forms. Its age is deliberated as Late Paleocene, notably excessively to Early Eocene. Even though it is prolonged, all the Paleocene in the Ziarat region, as well as the axial belt area where the Sangiali and Rakhi Gaj formations, i.e., the lower, along with the middle Sangiali group, is not promoted. For instance, the Ziarat Laterite exhibited K-T boundary is contacted through the Parh along the Dunghan formation [28].

### 3.4.3. Ranikot Formation

The Ranikot formation was subsequently nominated as the Ranikot fortress in the Laki range adjacent to Sindh. Furthermore, the Ranikot group is recognized as infra-Nummulitic. The Ranikot formation has been segregated into three formations, recognized as the Khadro, Bara (lower Ranikot sandstone), and Lakhra (upper Ranikot limestone) formations. The lower part has sandstone of brownish yellow color and shale along with limestone. Correspondingly, the lower Ranikot has diversified sandstone and shale along with gray to brown limestone, and the shale is also present in the Lakhra formation [29].

### 3.5. Cretaceous Stratigraphy

The Cretaceous stratigraphy of the area includes the following formations:

### 3.5.1. Sembar Formation

The name "Sembar formation" was introduced by Williams [30], after Sembar Pass in the Mari hills, to include the lower part of the "Belemnite beds" and "Belemnite shales". The "Belemnite beds" of Oldham also include the overlying Goru formation and the Parh limestone. According to Fatmi [31], the Sembar formation consists of black silty shale with interbeds of black siltstone and nodular rusty weathering argillaceous limestone beds or concretions. The upper contact is generally gradational with the Goru formation.

### 3.5.2. Goru Formation

The name "Goru formation" was introduced by Williams [30]. As described by Fatmi [31], the Goru formation consists of inter bedded limestone, shale, and siltstone.

The limestone is fine-grained, thin-bedded, and light to medium gray and olive gray. The interbedded shale and siltstone are gray, greenish gray, and locally maroon in color.

### 3.5.3. Parh Limestone

The term "Parh" was introduced by Blanford [32]. The Parh limestone, as described by Fatmi [31], is a lithologically very distinct unit. It is a hard, light gray, white, cream, olive green, thin-to medium-bedded, lithographic to porcellaneous, argillaceous and occasionally platy to slabby limestone with subordinate calcareous shale and marl intercalations. The formation is correlated with the Kawagarh formation of Kohat-Potwar province. The lower contact is with the Goru formation, while the upper contact is with the Moghal Kot formation.

### 3.5.4. Moghal Kot Formation

The term "Mughal Kot formation" was applied by Williams [30]. Lithologically, as stated by Fatmi [31], the formation comprises dark gray calcareous mudstone and calcareous shale with intercalations of quartos sandstone and light gray argillaceous limestone.

### 3.5.5. Pab Sandstone

The term "Pab sandstone" was introduced by Eschard et al. [33]. The formation typically consists of quartos sandstone, which is white, cream, or brown, weathering yellow brown, medium to coarse-grained thick bedded to massive, and shows cross-stratification. Some marl and argillaceous limestone, similar to that of the Parh limestone, are found intercalated.

### *3.6. Jurassic Stratigraphy*

The Jurassic stratigraphy of the area includes the following formations:

### 3.6.1. Samana Suk Formation

The name came from a mountain peak in the Samana range, which is revealed. It is constituted of limestone that is absolutely fine-grained and clay, along with sand. The lower association is with the Shinawri and topmost with the Chichali formation. The environment of precipitation is slightly nautical [34].

### 3.6.2. Shinawri Formation

It lies in Shinawri village in the western part of the Samana range. It has sparse to well-stratified limestone nodular marl, calcareous along with non-calcareous shale, and calcareous sandstone. Its lower association is conformable with the Datta and topmost with the Samana Suk formation. Its environment of precipitation is nautical [35].

### 3.6.3. Datta Formation

It is positioned in Datta Nala of the Surghar range. The lithology is constituted of arenaceous, siltstone, and shale. It has no peculiar petrified report; however, some carbonaceous remnants are present. Based on the law of superposition, its age is Early Jurassic. Datta shale acted as the superior originator rock and sandstone as the superior reservoir rock. Its environment of precipitation is deltaic [36].

### *3.7. Triassic Stratigraphy*

The Triassic stratigraphy of the area includes the following formations:

### 3.7.1. Kingriali Formation

It is positioned in Zaluch Nala, Tapan Wahan, at the Khisor range. It has two components, nominated as the Doya component, which is constituted of sandstone, dolomite, dolomitic sandstone, and shale, and the Vanjari component, which is mainly dolomitic. Its environment of precipitation is tidal flats [37].

### 3.7.2. Tredian Formation

It has also two components, Landa and Khatkiari. The Landa component's lithology is sandstone along with shale, while the Khatkiari component is constituted of compacted white-appearing sandstone. Its environment of precipitation is fluvial (non-marine) [38].

### *3.8. Permian Stratigraphy*

The Permian stratigraphy of the area includes the following formations:

### 3.8.1. Amb Formation

The Amb formation is positioned in Amb village. It has three units: the lower unit is constituted of shale, the intermediate is constituted of limestone, and the topmost is constituted of shale. As a result of the known index of petrified fusulinids, its age is nominated as Late Permian. Its lower association is conformable with the Sardhai and topmost with the Wargal formation. Its environment of precipitation is nautical [39].

### 3.8.2. Sardhai Formation

The Sardhai formation is positioned in Sardhai Gorge. It is constituted of clays of various appearances with a minute extent of sandstone and siltstone. The topmost part of Sardhai is calcareous. The limestone strata have well-improved petrification. Its environment of precipitation is fluvial delta. The age is Early-Middle Permian [40].

### 3.8.3. Warcha Formation

Its lithology is constituted of intermediate to coarse-grained zigzag stratified sandstone, conglomerate in places, and interbeds of shale are also present. The pebbles of the unit are particularly of granite of pink appearance along with quartzite. Based on the law of superposition, its age is Early Permian [41].

All of these formations overlay the Salt Range formation, Khewra sandstone, and Kussak formation (composed of sandstone and glauconitic shale), respectively, as shown in Figure 1c.

## 4. Seismic Data Interpretation

Seismic interpretation is the description of the subsurface geological features from seismic data collected during the extraction process. The drilling team can perform the interpretation of seismic data to determine the structural models beneath the earth's surface. Amplitude, frequency, and phase are the main seismic attributes that are generally used in the interpretation process.

The main aim of interpretation is to identify the different reflectors or horizons as interfaces between different geological formations. Interpretation is mainly required in order to obtain structural and stratigraphic information. During interpretation, the horizons and faults are marked for better representation. In the beginning, interpretation was done manually on paper sections, but nowadays, it is done with the help of powerful computer systems. With graphic support, computer-aided interpretation systems are used in the industry, as shown in Figure 1d. Seismic sections can forecast configurations of up to tens of kilometers. A fault with throw that is less than one-quarter of the wavelength of the seismic wave will be very complicated to mark out in the seismic section [42]. The current study area is situated in an extensional regime, so the general structure is normal related, i.e., horst and graben structure. A normal fault will progress due to the extensional regime. Overall, four faults are apparent on the seismic section, which increases the complexity of the research area. These are marked upon the detection of abrupt variations in the condition of reflectors and misrepresentation or disappearance of the reflection underneath the faults.

The purpose of interpretation is to understand how these structures are formed, and it is necessary for mapping and marking the horizons above and below the target zone. The strata in an exposure or outcropping of sedimentary rock can range from layers as

thin as paper, which are known as lamina (plural: laminae or laminations), to beds tens of feet thick. Generally, the more stable and consistent the environmental conditions during deposition, the thicker the strata will be.

The seismic interpretation procedure consists of various tracks reserved to find an explanation of the seismic sector on the subject of all stratigraphic structures and sequences [13]. In the oil and gas industries, the critical objectives of seismic interpretation are to detect hydrocarbon pathways and explore its flow [14]. In the case of the Kabirwala lines, the reflectors are constituted to be fundamentally flat; thus, a conduit cannot be assembled for hydrocarbon migration along with maturation. Seismic interpretation is also necessary for selecting the appropriate geological reservoir for long-term $CO_2$ sequestration [43–49].

The basic interpretation stages are based on seismic well log data of the Tola-01 well in the Kabirwala area and on seismic lines. As given in Tables 2 and 3, five migrated seismic lines, navigation data of seismic lines, well log data, and formation tops for the Tola-01 well were accessible for this project. Seismic data analysis consists of stratigraphic and structural analysis. The structure and stratigraphy of the subsurface is evaluated from the seismic reflection data [50]. Seismic interpretation involves converting velocity and time into the depth of subsurface reflecting interfaces to translate seismic data into geological images [5].

**Table 2.** Seismic line information.

| Sr. No. | Seismic Line | Line Style | Direction |
|:---:|:---:|:---:|:---:|
| 1 | 875-KBR-230 | Strike | N-S |
| 2 | 875-KBR-229 | Strike | N-S |
| 3 | 875-KBR-231 | Dip | E-W |
| 4 | 875-KBR-221 | Dip | E-W |
| 5 | 875-KBR-220 | Dip | E-W |

**Table 3.** Well log data of the study area.

| Sr. No. | Tola-01 | Unit |
|:---:|:---:|:---:|
| 1 | Sonic (DT) | μsec/ft |
| 2 | Neutron porosity (NPHI) | Fraction |
| 3 | Formation density (RHOB) | $gm/cm^3$ |
| 4 | Resistivity (MSFL) (LLD) (LLS) | Ω.m |
| 5 | Gamma ray (GR) | API |
| 6 | Spontaneous potential (SP) | mV |
| 7 | Caliper (CALI) | Inches |

Figure 2a points out the "Base Map" of the study area (Tola-01 well), which is located in the Punjab platform of Central Indus Basin (CIB). The key purpose of the "Base Map" is to point out the particular direction of the seismic lines, shot points, as well as the well position on the seismic line. In the current study, the "Base Map" points out two strike lines and three dip lines. The two strike lines have the shooting trend from the north to south direction. However, the three dip lines have the shooting trend from the east to west direction. The location of the Tola (1) well is added in the base map. The Tola (1) well is located at 875-KBR-230 line in the Base Map. Owing to the seismic data interpretation, four reflectors are defined in the seismic sections. The well tops of Tola-01 are utilized for the correlation. The horizons are the Dunghan, Samana Suk, Datta, and Warcha formations, as shown in Table 4. Similarly, the horizons of seismic lines are given in detail in Figures 3–7. The Tola (01) well is located on the 875-KBR-230 line in the base map. The 875-KBR-230 line is used as a reference for marking our seismic sections. The well tops of the Tola-01 well were used to locate the required reflectors on the seismic section. The well tops data give us the formation tops with a required depth. As far as the well tops of the Tola-01 well

are concerned, the tops of the required formations, which are Dunghan, Samana suk, Datta and Warcha, are given in Tables 4 and 5.

The fit line curve created from the velocity windows calculation and the graph help us find the time against the depth on the time section. The time values of the prescribed formations have been obtained using the time depth chart. Then, these time values obtained from the time-depth chart of well Tola-01 were projected on the seismic line 844-KBR-230. Furthermore, a line is drawn from the prescribed shot point crossways on the seismic traces toward the second time vertically, and the values of the time are also pinpointed on the well line over the seismic line. The time picked on the seismic line is shown in Tables 4 and 5. For labeling the seismic horizons and faults, the time of each reflector will be recognized at various shot points, and the travel time of each reflector will be marked along with faults on the *y*-axis, in contrast to the shot points on the *x*-axis.

In the present study, there are four marked reflectors with normal faults on each line. Gridding of these lines was assembled for a contouring plan as well as to interpret the configuration model of these lines for the arrangement of contouring. The horst along with the graben configurations that existed on the seismic sections (Figures 3–7) may be a proper vicinity for the gathering of hydrocarbons. Thus, the labeled seismic sections are the promoted sections of the reflectors, which point out the subsurface configurations in the time domain. The values of the time depth are taken on the seismic line 845-KBR-45. Therefore, the trace on a specific time, using the referred time scaling, is used to locate the time on the well spot. In addition, after four traces are marked, four of them are therefore spread, using the common line joining the trace. The trace is observed by watching it parallel to the paper orientation rather looking from the top. Then, the horizons are marked on the seismic line. The time contour maps of four horizons are organized and demonstrated in Figures 8a, 9a, 10a and 11a. The average stacking velocities are used to convert the two-way travel time (TWT) of these four horizons into depth at Tola-01, and the resulting depth structure maps are shown in Figures 12a, 13a, 14a and 15a. The seismic line 844-KBR-5, which is a strike line, was reserved as a control line for the correlation.

### 4.1. Seismic Time and Depth Sections

The first step is to solve the velocity windows, which are given on the top of the seismic lines. As shown in the Base Map in Figure 2a, the Tola-01 well lies on the seismic line 875-KBR-230; therefore, the velocity window is solved for the KBR-230 seismic section. On the velocity window, one column is for the time in milliseconds, and the other column is for the RMS (root-mean-square) velocity. Then, the RMS velocity is multiplied with the one-way travel time in milliseconds to give the depth by using the formula S = VT. The one-way travel time is calculated by dividing the time by 2. This column gives the depth of each time during the recording of a seismic wave traveling under the surface. For more insight into the velocity window, the depth needs to be plotted against the time window [51,52]. The velocity windows are first solved to obtain the time values, and then, the depth is plotted on the horizontal axis against the time on the vertical axis. By using the reasoning that the subsurface structures can be predicted by transforming the time into depth, the TVT chart is constructed. The perception of transforming time to depth is accurate to point out the subsurface structures because the structure is flat and the map is in terms of the two-way travel time (TWT) of sound waves. Thus, a map of the seismic time section is proposed to point out the structure in the subsurface. The depth is plotted with time, as shown in Figure 2b.

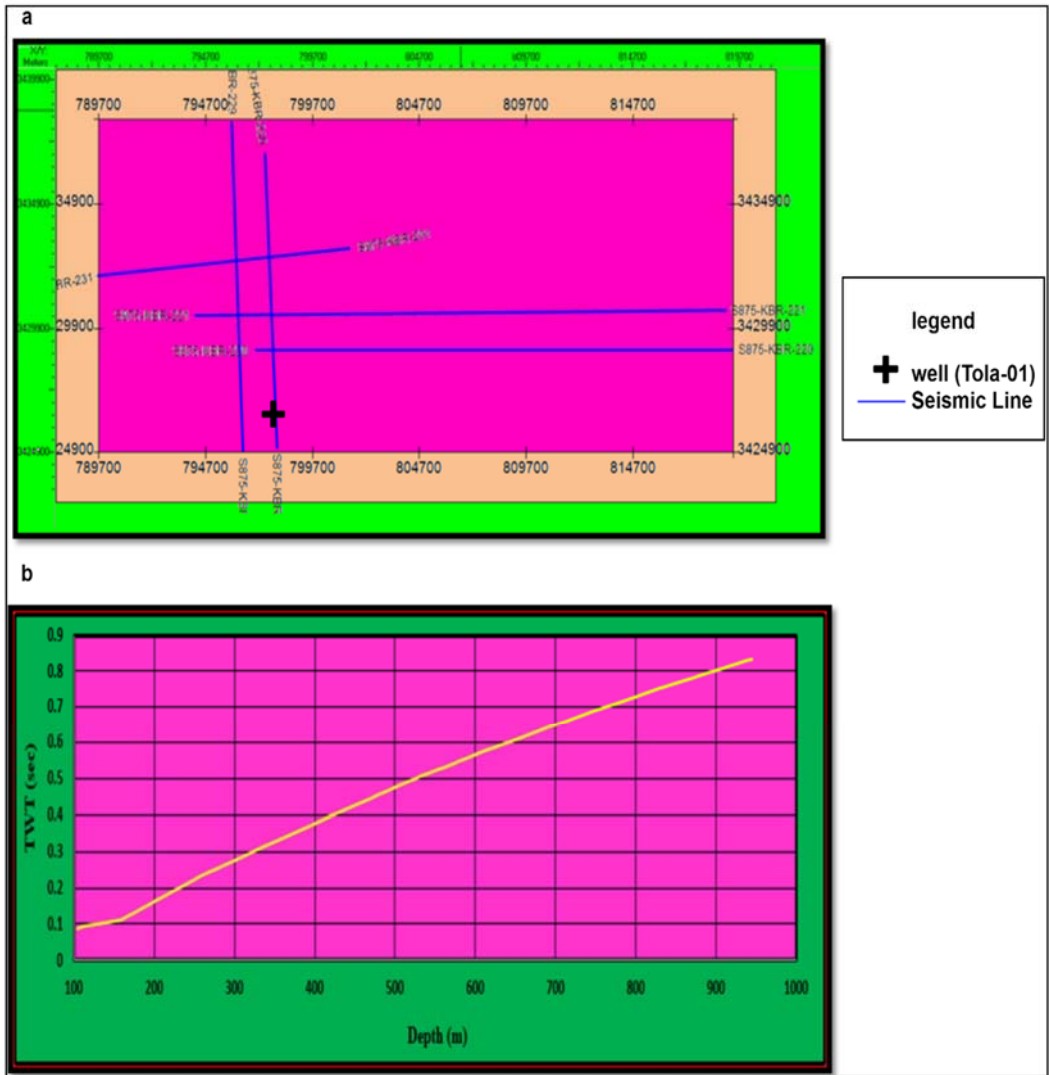

**Figure 2.** (**a**) Base map of the study area. (**b**) Time-depth (TD) chart of the study area.

**Table 4.** Formation top values of marked reflectors.

| Formation Name | Depth |
|---|---|
| Dunghan | 1080.46 m |
| Samana Suk | 1174.05 m |
| Datta | 1400 m |
| Warcha sandstone | 1810 m |

**Table 5.** Marked horizons obtained from time-depth chart estimated in Tola-01 well.

| Formation Name | Time |
|---|---|
| Dunghan | 0.9 s |
| Samana Suk | 0.96 s |
| Datta | 1.08 s |
| Warcha sandstone | 1.24 s |

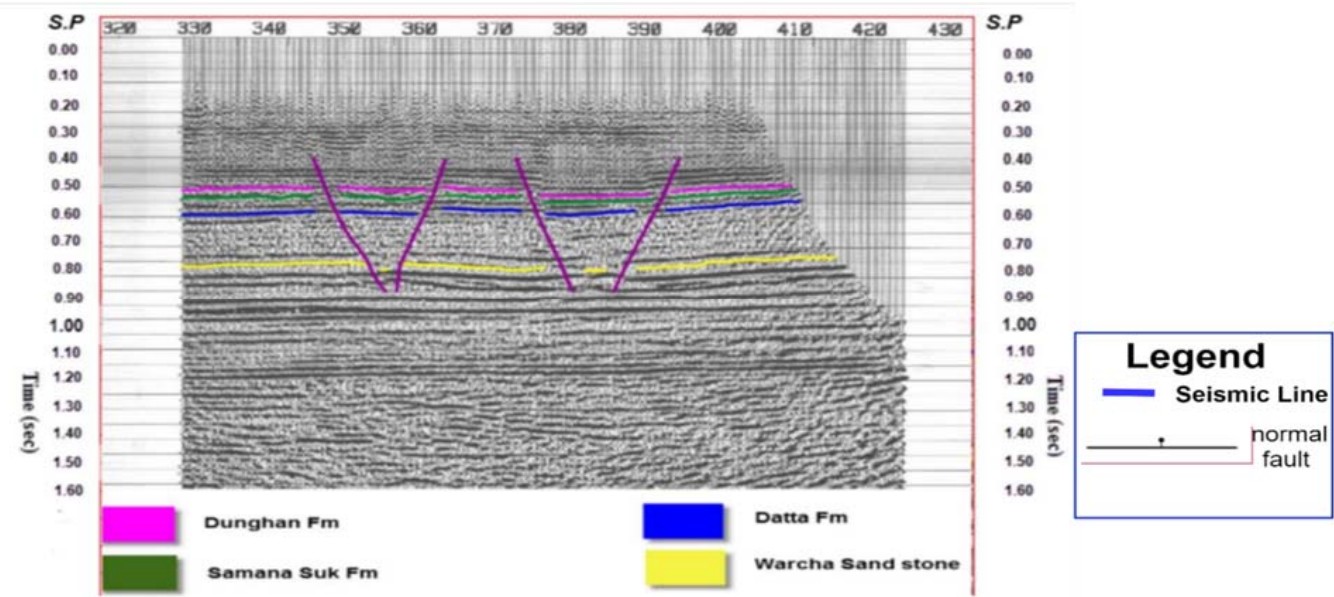

**Figure 3.** Marked seismic horizons of 875-KBR-220 line.

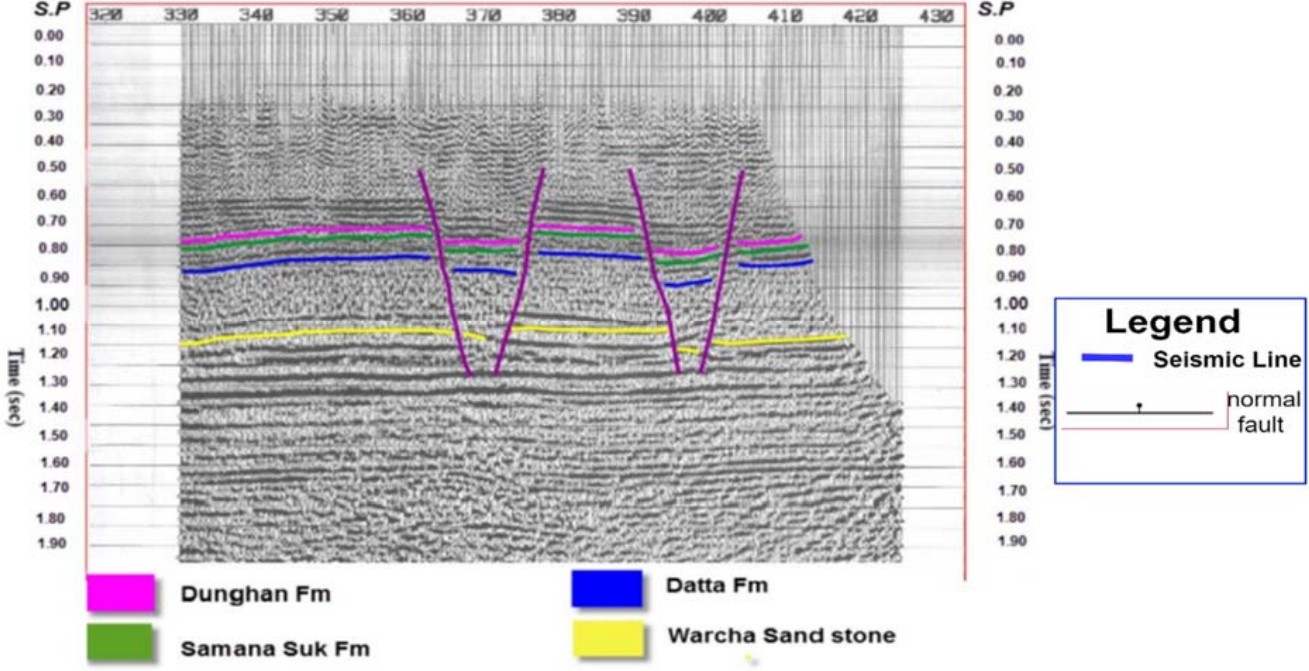

**Figure 4.** Marked seismic horizons of 875-KBR-221 line.

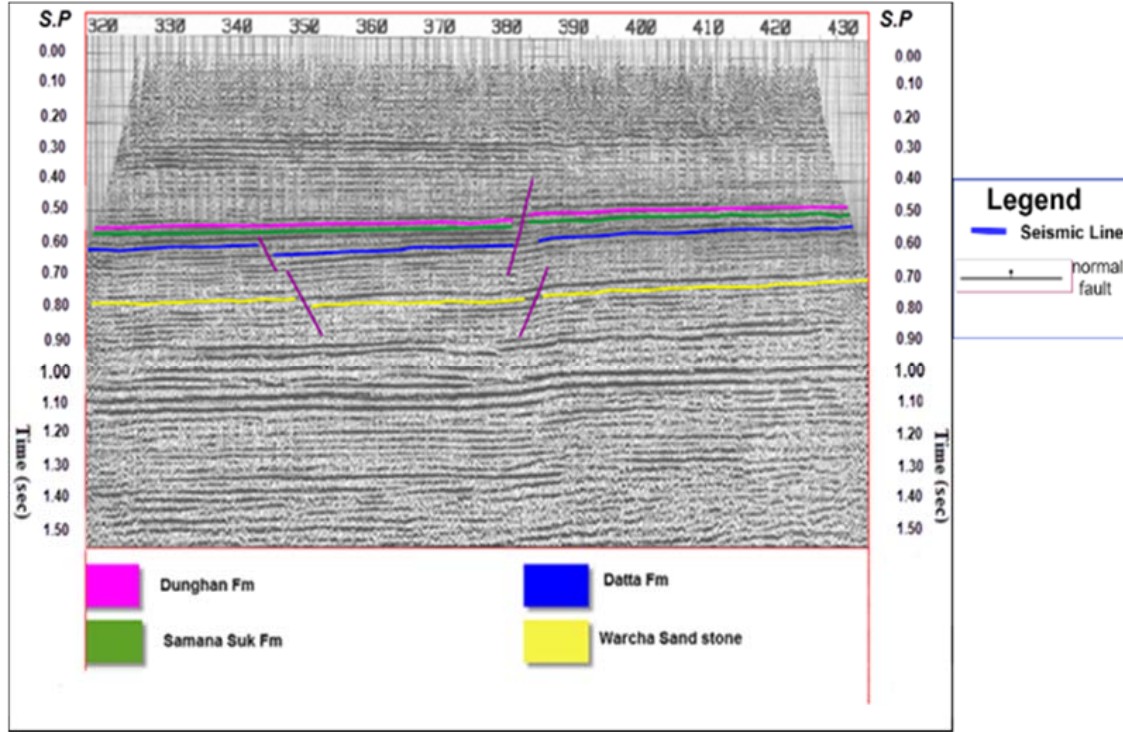

**Figure 5.** Marked seismic horizons of 875-KBR-229 line.

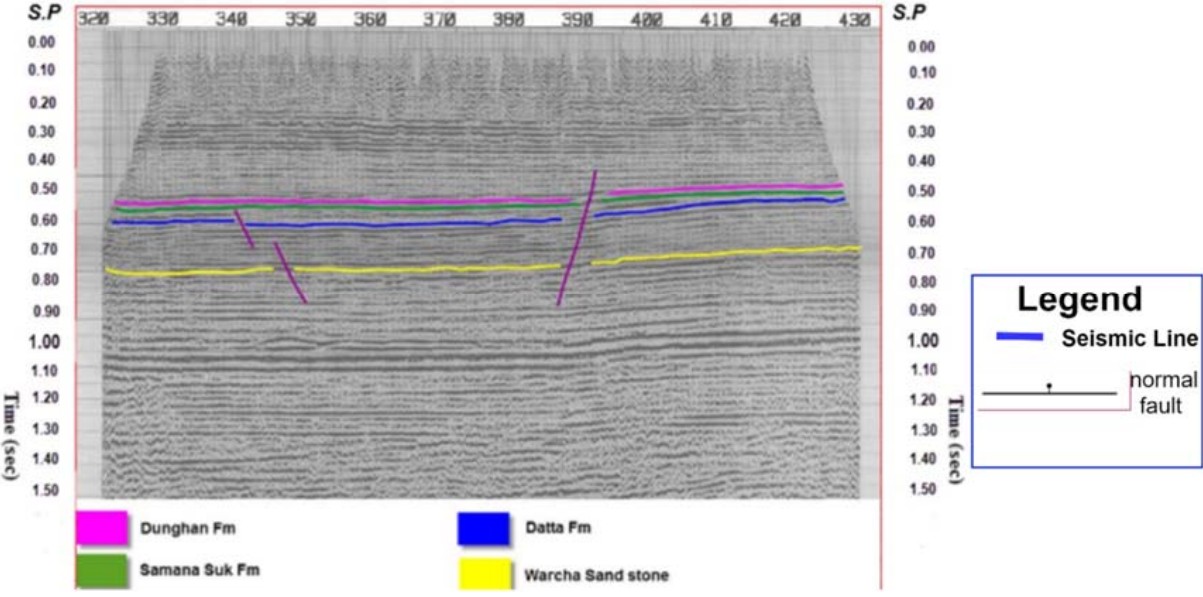

**Figure 6.** Marked seismic horizons of 875-KBR-230 line.

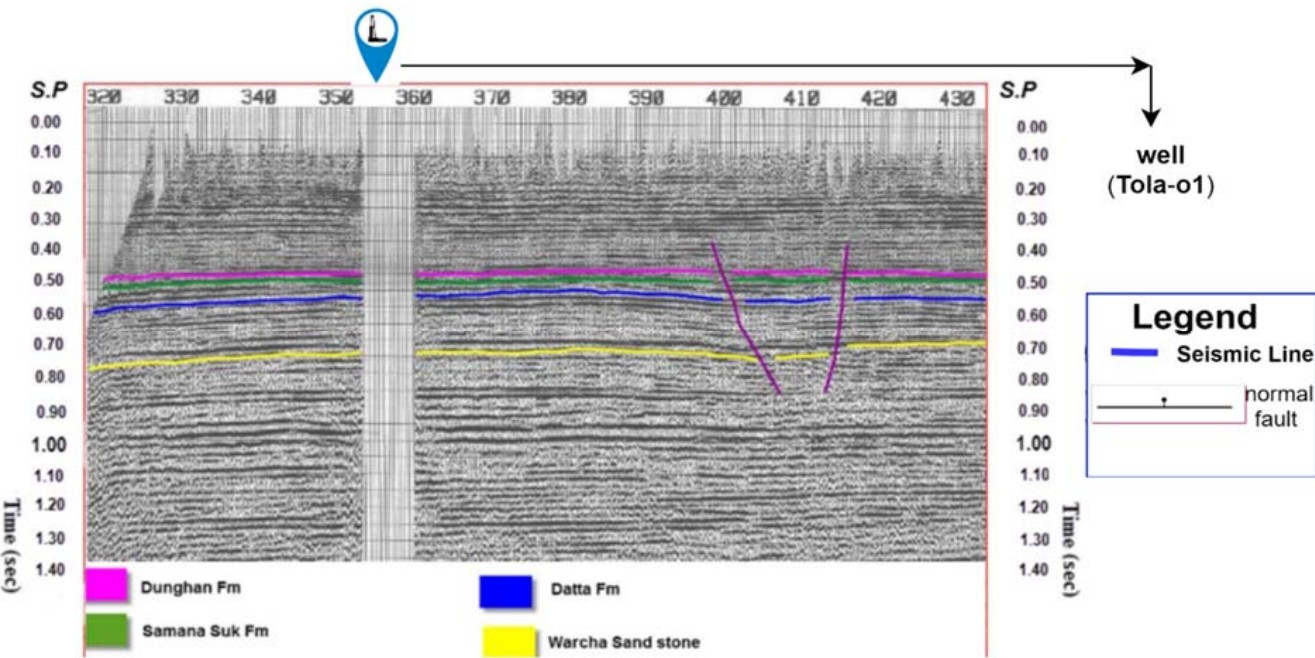

**Figure 7.** Marked seismic horizons of 875-KBR-231 line.

### 4.2. Marking Reflectors

To interpret the seismic sections, it is important to point out a particular well on a specific line to be used as a reference [53]. In this study, the seismic line and the well tops of the Tola-01 well were used to pinpoint the prescribed reflectors on that particular seismic section. Then, the nature of the marked reflectors is traced toward the seismic section used in this study. The well tops of Tola-01 of the prescribed formation are Dunghan, Samana Suk, Datta, and Warcha formations. The Punjab platform is tectonically less influenced area because of its greater outdistance from the belt, as shown in Tables 4 and 5.

### 4.3. Time and Depth Contour Maps and Surfaces Maps

A mark that joins the track of equivalent values is called a contour mark or line. Such drawings display the sharpness of angles (slope), the promotion top of the subsurface of the deposited rock stratum (elevation), and the corresponding two-way travel time of the prospect in milliseconds [54].

#### 4.3.1. Time Contour Maps

A contour map is simply defined as a line that exhibits parameters on a 2D map, such as depth or elevation [20]. With the help of time rates, the contours are created to show the time of the waves that reflect back from pronounced condensed reflectors or strata due to the density/velocity contrast or simply acoustic impedance. The time contour is initiated on a map for every reflector. In our research, the reflectors of relevance are the Dunghan, Samana Suk, Datta, and Warcha formations. The time contour maps of the four essential reflectors on five seismic sections are projected on the Base Map. Justified to the depth along with other diverging capacities in the path of the traveling waves, the reflection time is further computed in order to give an idea about the structures and the time delay of the seismic wave feedback.

#### 4.3.2. Depth Contour Maps

A depth contour map points out the structures of the area and contributes to the overall information about the depth along with its range of precise formation [21]. The given depth contour maps are constructed with the help of the reflection time rates of

the seismic lines of the Kabirwala area, which contains 875-KBR 231, 875-KBR-230, 875-KBR-229, 875-KBR-221, and 875-KBR-220. On the seismic section, there are four reflectors: Dunghan, Samana Suk, Datta, and Warcha formations. It is assured by the contour maps that the reflectors are laterally flat, which can be seen in their patterns. These reflectors are marked as T1, T2, T3, and T4 on the seismic sections along with the depth contour graphs of these reflectors. For a single depth value, the depth contour map points out the same depth contour line, and the contour contributes to help us build a three-dimensional structure of a horizon showing its depth as a surface.

### 4.4. Time Contour and Time Surfaces Maps of Dunghan Formation

The time contour map of the Dunghan formation points out a monocline structure dipping in the eastward orientation. The reflection time increases from 0.93 to 1.14 s, as shown in Figure 8a. The time contour surface of the Dunghan formation was generated by joining the points of equal time. These time values were assigned a color bar, which can be used as a guide for interpretation. On the color bar, red shows the shallowest point, and purple-blue shows the deepest point (Figure 8b). The 3D view of the particular formation surface helps to comprehend the monocline structure in the Punjab platform region northward.

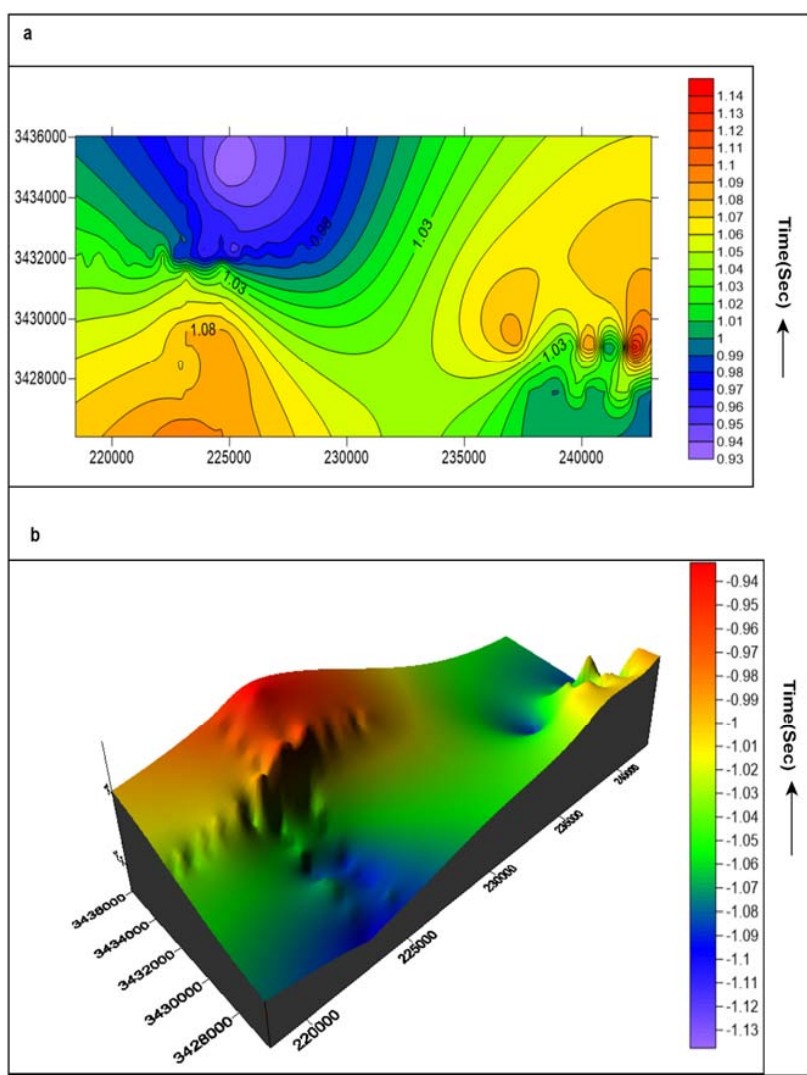

**Figure 8.** (**a**) Time contour map of the Dunghan formation, (**b**) Time surface map of the Dunghan formation.

The counter map is generally defined as the line showing the same parameter (depth elevation) on a 2D map. The contours are made with the help of the time values, which represent the time of the wave to reflect from a dense point reflector such as R1, R2, R3, and R4. The time counters are generated on the map, taking the value of reflector time filling a line. In the current research, the reflectors of interest are Dubghan, Samana Suk, Datta, and Warcha formations, and these are plotted on the counter section referring to their reflection time values. The reflection time is plotted in order to give an idea about the structure and time delay of the seismic wave response due to the depth and other deflections in the path of the wave.

### 4.5. Time Contour and Time Surfaces Maps of Samana Suk Formation

The time contour map of the Samana Suk formation also points toward the monocline structure dipping moderately in the eastward direction. The increase in reflection time is from 0.97 to 1.19 s, which is shown in Figure 9a. The time contour surface of the Samana Suk formation was generated by joining the points of equal time. These time values were assigned a color bar, which can be used as a guide for interpretation. On the color bar, red shows the shallowest point and purple-blue shows the deepest point (Figure 9b).

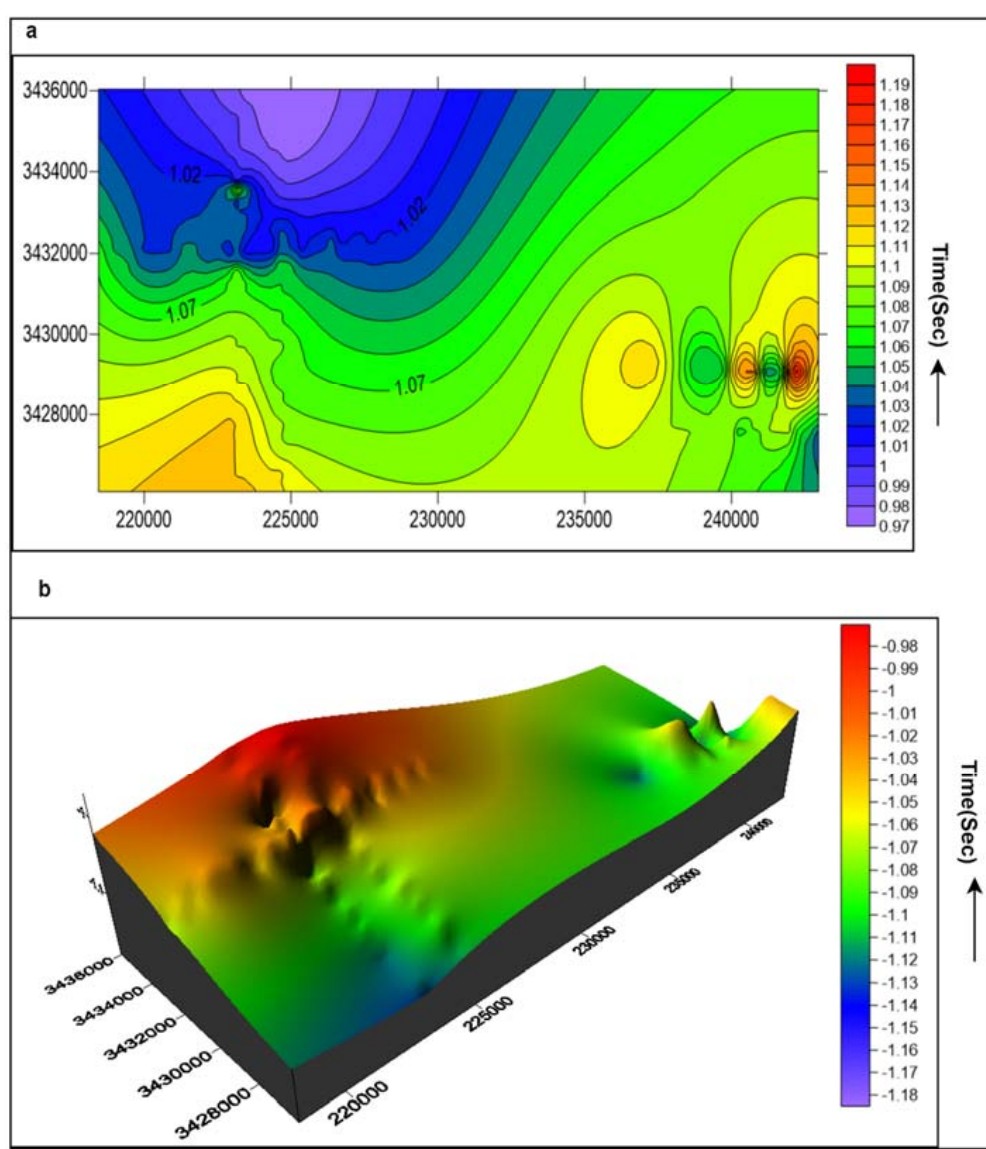

**Figure 9.** (**a**) Time contour map of the Samana Suk formation, (**b**) Time surface map of Samana Suk formation.

### 4.6. Time Contour and Time Surfaces Maps of Datta Formation

The time contour map of the Datta formation also has a monocline structure, similar to the other two formations, describing the characteristics of the Punjab platform. The increase in reflection time is from 1.02 to 1.28 sec, which is shown in Figure 10a. The time contour surface of the Datta formation was generated by joining the points of equal time. These time values were assigned a color bar, which can be used as a guide for interpretation. On the color bar, red shows the shallowest point and purple-blue shows the deepest point (Figure 10b).

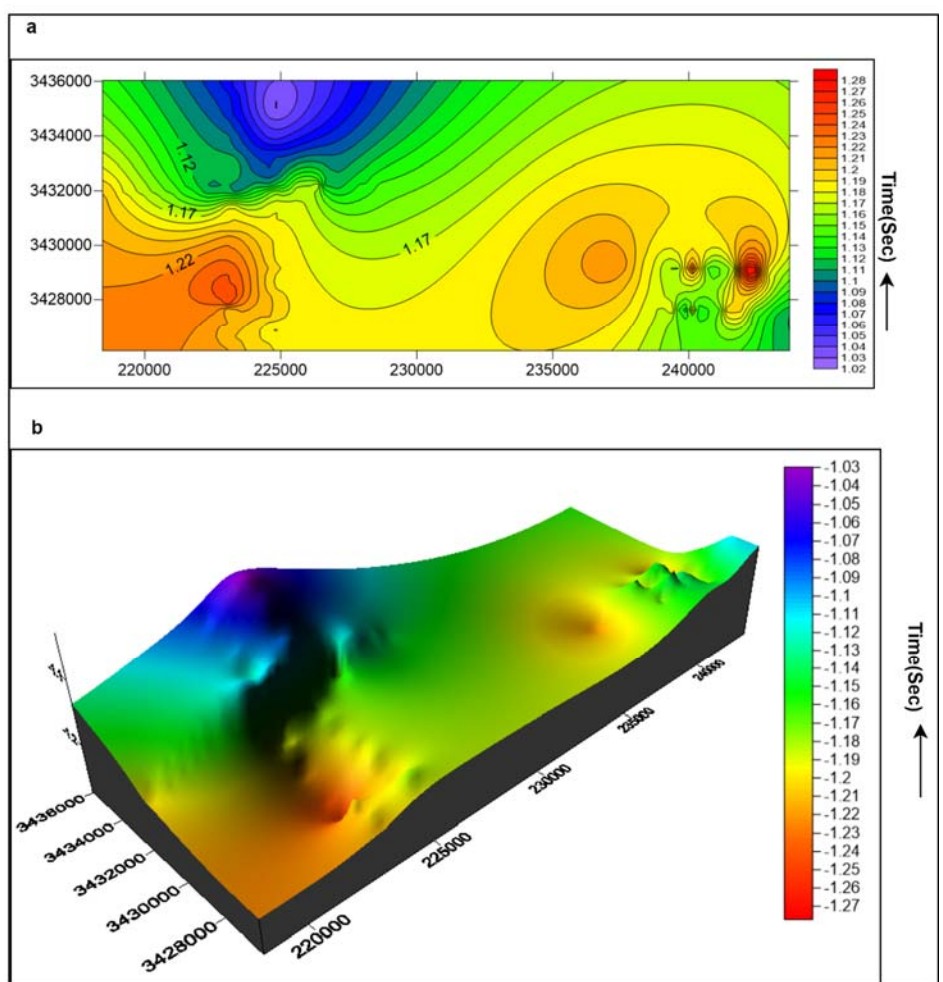

**Figure 10.** (**a**) Time contour map of the Datta formation. (**b**) Time surface map of Datta formation.

### 4.7. Time Contour and Time Surfaces Maps of Warcha Formation

The time contour map of the Warcha formation also points toward the monocline structure, dipping moderately in the eastward direction. The increase in reflection time is from 1.34 to 1.64 s, as shown in Figure 11a. The time contour surface of the Warcha formation was generated by joining the points of equal time. These time values were assigned a color bar, which can be used as a guide for interpretation. On the color bar, red shows the shallowest point and purple-blue shows the deepest point (Figure 11b).

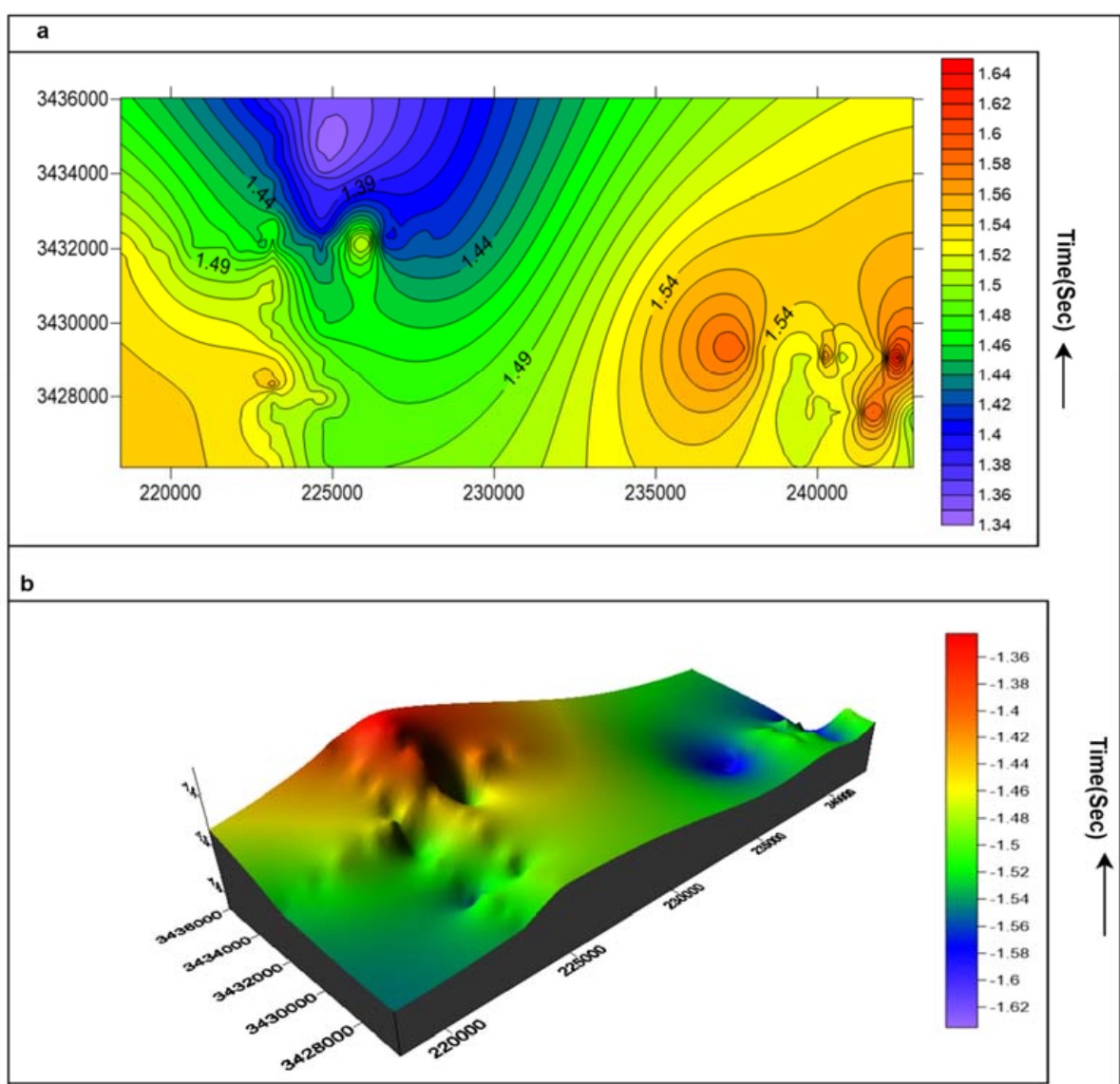

**Figure 11.** (**a**) Time contour map of the Warcha formation. (**b**) Time surface map of the Warcha formation.

### 4.8. Depth Contour and Depth Surfaces Maps of Dunghan Formation

In comparison to the time contour map, the depth contour map of the Dunghan formation contributes to the same diagram of the dipping monocline structure of the Punjab platform area of the CIB. The depth value of the Dunghan formation plunges eastward and the decrease in depth value is from 1370 to 1110 m, as shown in Figure 12a. The depth contour surface of the Dunghan formation was generated by joining the points of equal depth. These depth values were assigned a color bar, which can be used as a guide for interpretation. On the color bar, red shows the shallowest point and purple-blue shows the deepest point (Figure 12b).

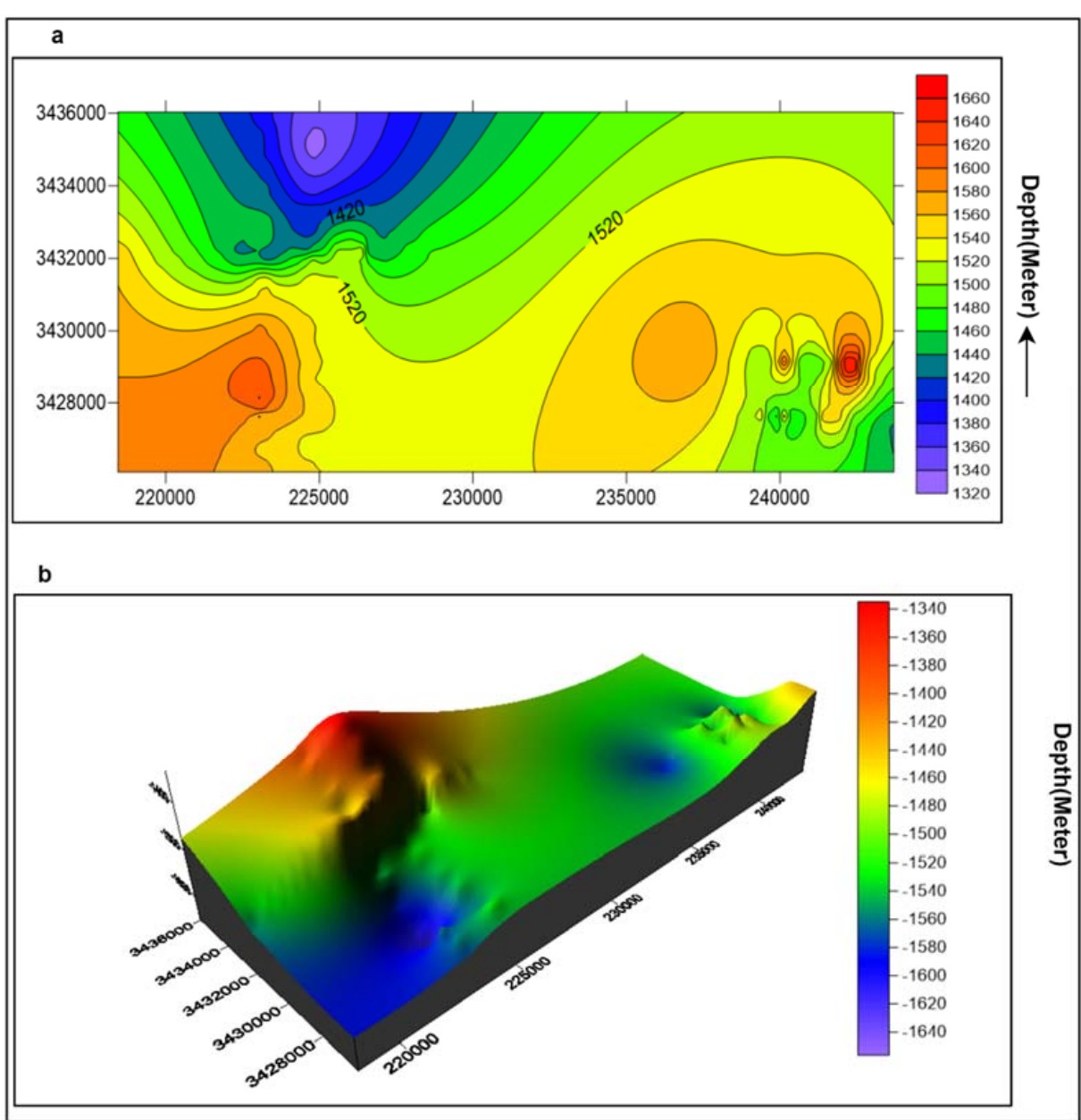

**Figure 12.** (**a**) Depth contour map of the Dunghan formation. (**b**) Depth surface map of the Dunghan formation.

### 4.9. Depth Contour and Depth Surfaces Maps of Samana Suk Formation

The depth contour map of the Samana Suk formation points out descending depth northward from 1450 to 1180 m, as shown in Figure 13a. The depth contour surface of the Samana Suk formation was generated by joining the points of equal depth. These depth values were assigned a color bar, which can be used as a guide for interpretation. On the color bar, red shows the shallowest point and purple-blue shows the deepest point (Figure 13b).

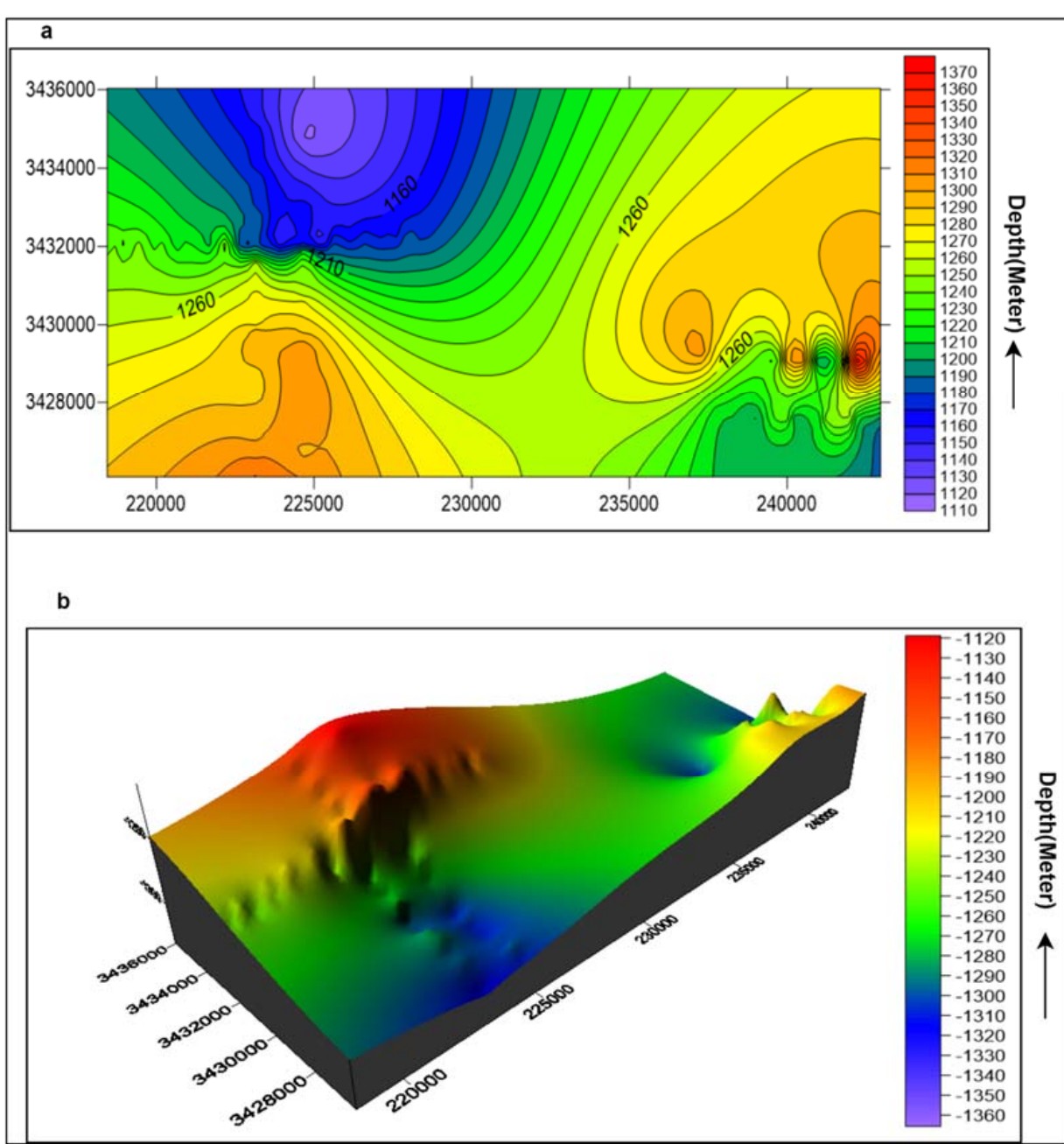

**Figure 13.** (**a**) Depth contour map of the Samana Suk formation. (**b**) Depth surface map of the Samana Suk formation.

*4.10. Depth Contour and Depth Surfaces Maps of Datta Formation*

　　　　The depth contour map of the Datta formation uses a similar template as the Samana Suk formation. The inclusive structure is the monocline structure of the Punjab platform, even though fluctuating variations in depth are shown. The decrease in depth value is from 1660 to 1320 m, as shown in Figure 14a. The depth contour surface of the Datta formation was generated by joining the points of equal depth. These depth values were assigned a color bar, which can be used as a guide for interpretation. On the color bar, red shows the shallowest point and purple-blue shows the deepest point (Figure 14b).

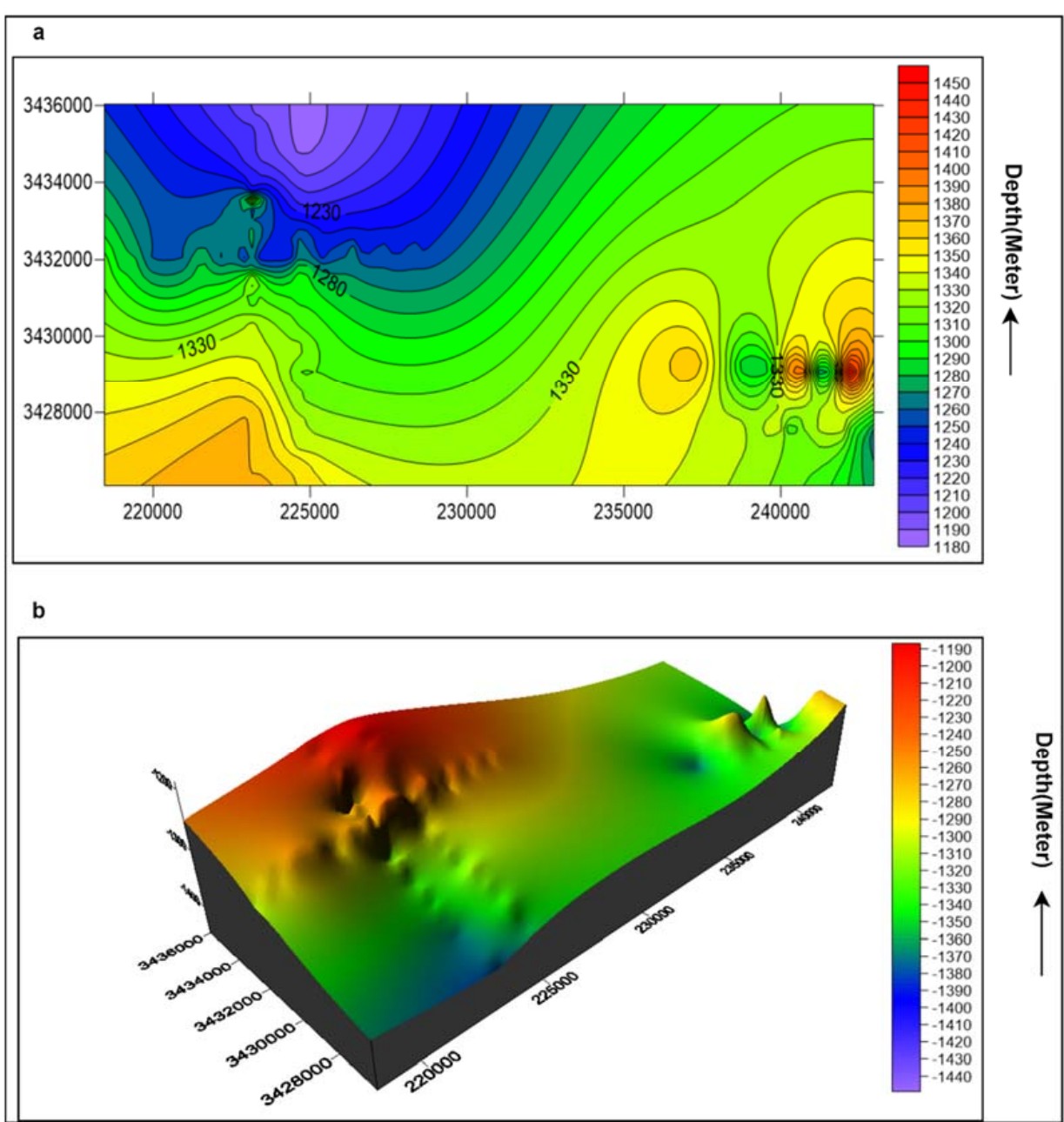

**Figure 14.** (**a**) Depth contour map of the Datta formation. (**b**) Depth surface map of the Datta formation.

### 4.11. Depth Contour and Depth Surfaces Maps of Warcha Formation

The depth contour map of the Warcha formation uses a similar template as the Datta formation. The inclusive structure is the monocline structure of the Punjab platform, even though fluctuating variations in depth are shown. The decrease in depth value is from 2400 to 1940 m, as shown in Figure 15a. The depth contour surface of the Warcha formation was generated by joining the points of equal depth. These depth values were assigned a color bar, which can be used as a guide for interpretation. On the color bar, red shows the shallowest point and purple-blue shows the deepest point (Figure 15b).

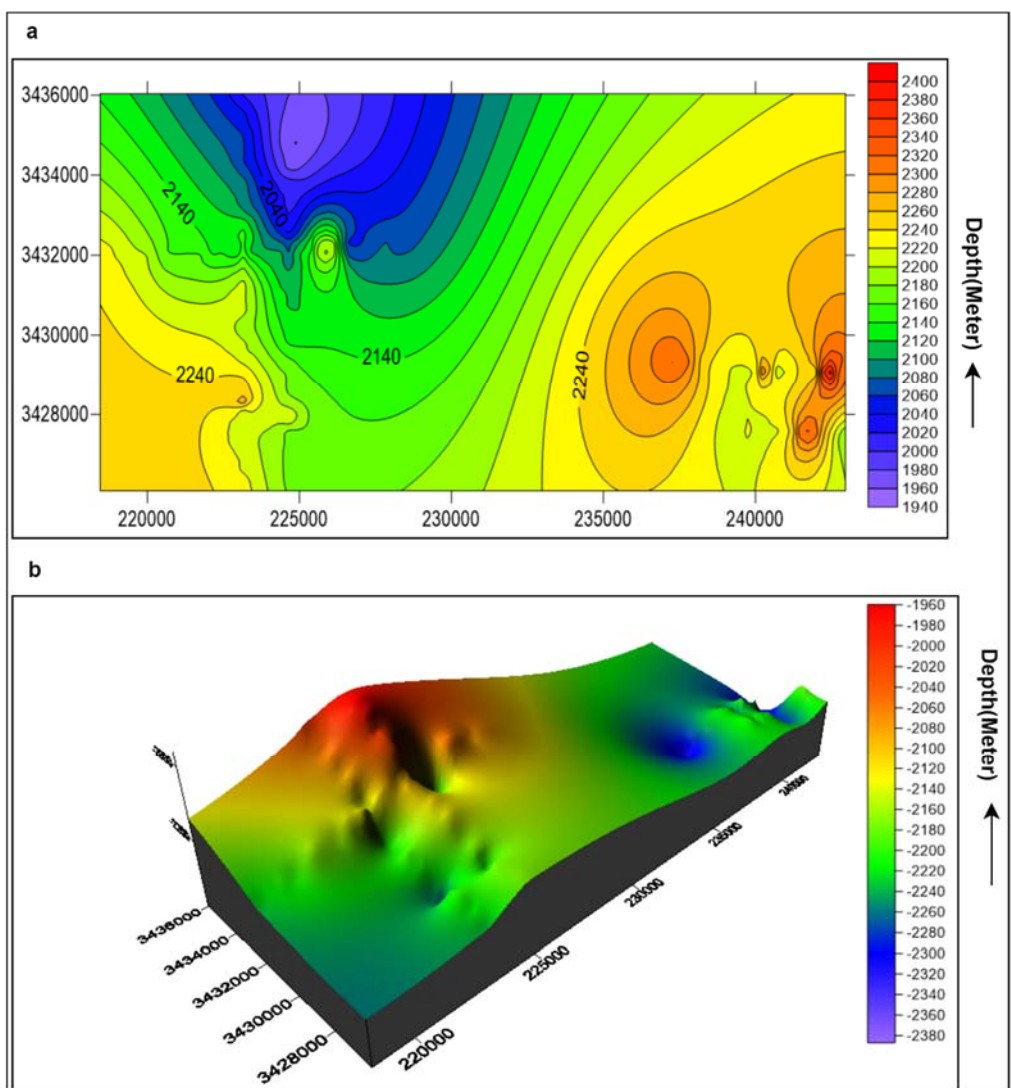

**Figure 15.** (**a**) Depth contour map of the Warcha formation. (**b**) Depth surface map of the Warcha formation.

## 5. Petrophysical Analysis

The main purpose of the petrophysical analysis is to retrieve information and data from the well logs [7,55,56]. Data utilized in the current research study include log curves of gamma ray, neutron, density, and resistivity logs. The clean zone is spotted by means of the gamma ray log. The separation in the resistivity curves is LLS (Letro-Log Shallow) MSFL (Micro Spherically Focused Log) and LLD (Letro-Log Deep) and the highest values of resistivity indicate the occurrence of hydrocarbons. The crossover between the neutron and density logs indicates the existence of hydrocarbons in the chosen clean zone. The subsequent zone of concentration is marked in Table 6, and the petrophysical parameters are shown in Table 7.

**Table 6.** Marked zones of interest encountered in Tola-01 well.

| Formation Name | Depth |
|----------------|-------|
| Dunghan | 1080.46 m |
| Samana Suk | 1174.05 m |
| Datta | 1400 m |
| Warcha | 1810 m |

**Table 7.** Petrophysical properties and their average values.

| Petrophysical Property | Average Value |
|---|---|
| Volume of shale | 69% |
| Average porosity (PHIT) | 27.8% |
| Effective porosity (PHIE) | 8.2% |
| Water saturation (SW) | 18% |
| Density porosity | 27% |
| Hydrocarbon saturation (SH) | 82% |

### 5.1. Calculation of Volume of Shale

The volume of shale (Vsh) is also called the dirtiness of the reservoir. Figure 16 shows the Vsh considered in the zone of interest. The following equation was used for Vsh calculation:

$$Vsh = (GR\ log - GR\ min)/(GR\ max - GR\ min) \tag{1}$$

where GR log is the gamma ray log reading, GR max is the maximum gamma ray value, and GR min is the minimum gamma ray value.

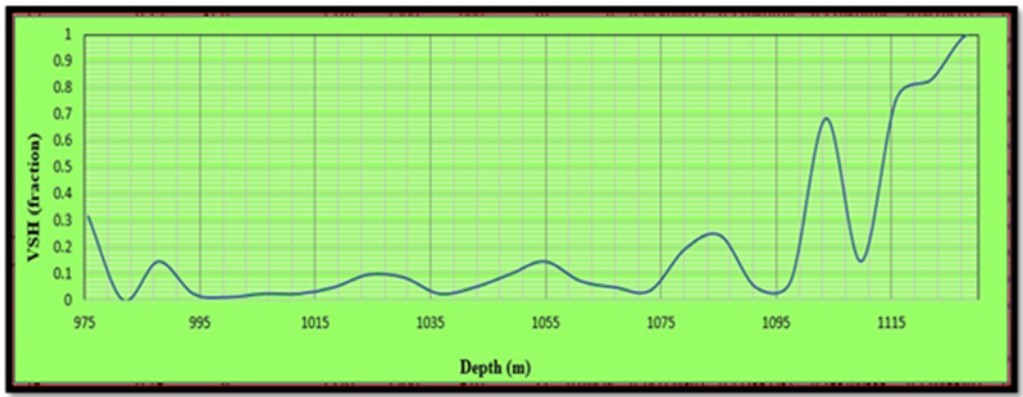

**Figure 16.** Volume of shale for selected zone.

### 5.2. Total Porosity

Basically, porosity is described as the quantity of void places (pores) relative to the overall volume of a rock [57]. These pores were formed in the rock during its deposition, which is termed as primary porosity, in contrast to the disintegration of particles through water or cracking, which is termed as secondary porosity. It is signified by $\phi$ and is often articulated in percentage. Porosity can be calculated using dissimilar techniques. In the current study, porosity was found using two methods [58]: density and neutron logs.

$$Total\ Porosity = (Density\ Porosity + Neutron\ Porosity)/2 \tag{2}$$

Total porosity (PHIT) is shown in Figure 17.

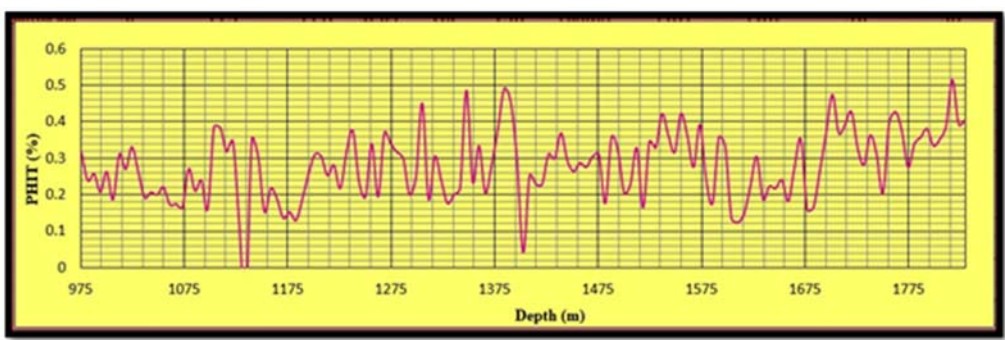

**Figure 17.** Total porosity variation along the depth.

### 5.3. Neutron Porosity

Neutron porosity records porosity, based on the influence of the accumulation on fast neutrons emanated by means of a source [59]. Hydrogen has an extreme impact on decelerating and catching neutrons. Hydrogen often resides in pore liquid, and the neutron porosity record reacts predominantly to porosity. The neutron porosity log is shown in Figure 18.

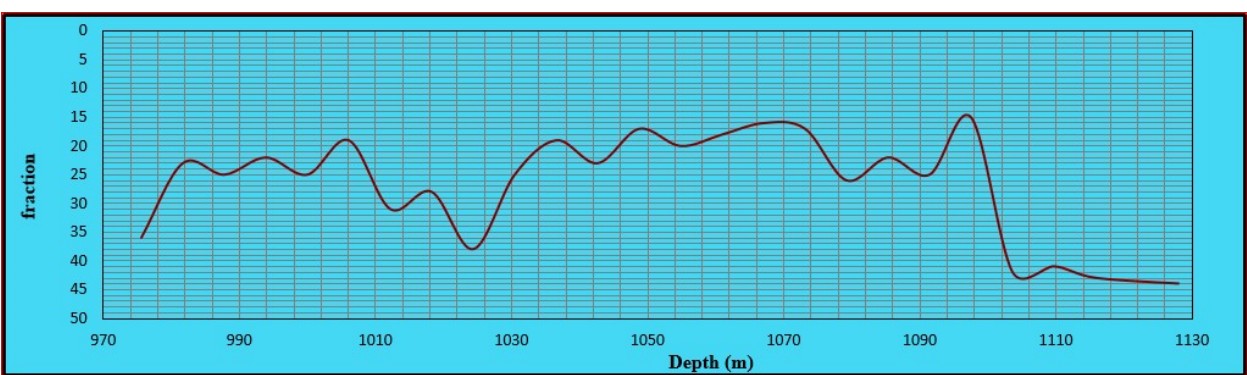

**Figure 18.** Neutron porosity variation along the depth.

Density Porosity

The density record is converted to porosity through the following relation:

$$\Phi_D = \rho_{ma} - \rho_b / \rho_{ma} - \rho_{fl} \tag{3}$$

where $\Phi_D$ is density porosity, $\rho_{ma}$ is matrix density (g/cm$^3$), $\rho_b$ is formation bulk density (log value) (g/cm$^3$), and $\rho_{fl}$ is the density of the fluid saturating the rock adjacent to the drill-hole, which is frequently mud filtrate (g/cm$^3$).

Incorrect values may perhaps result when conducting the analysis in evaporates or gas accumulations. They generate lower porosity values than expected authentic values. The density porosity log is shown in Figure 19.

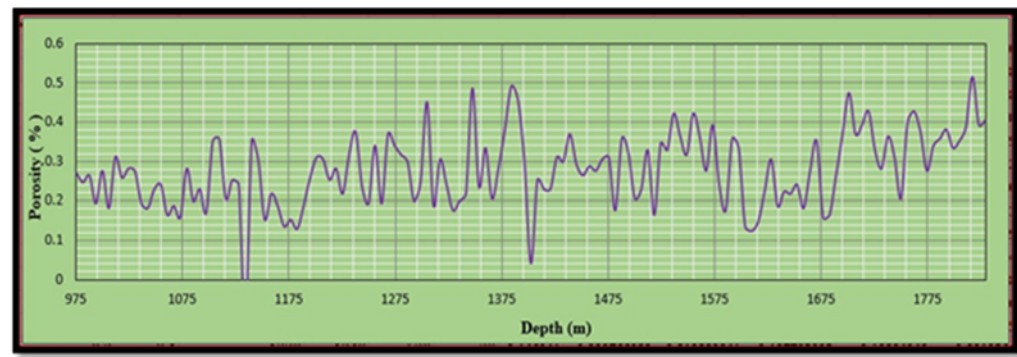

**Figure 19.** Density porosity variation along the depth.

### 5.4. Effective Porosity

Effective porosity (PHIE) is the summation of totally connected pores [7]. It is calculated by the following formula:

$$\text{Effective Porosity} = \text{Total Porosity} \times (1 - \text{Vsh}). \tag{4}$$

The effective porosity log is shown in Figure 20.

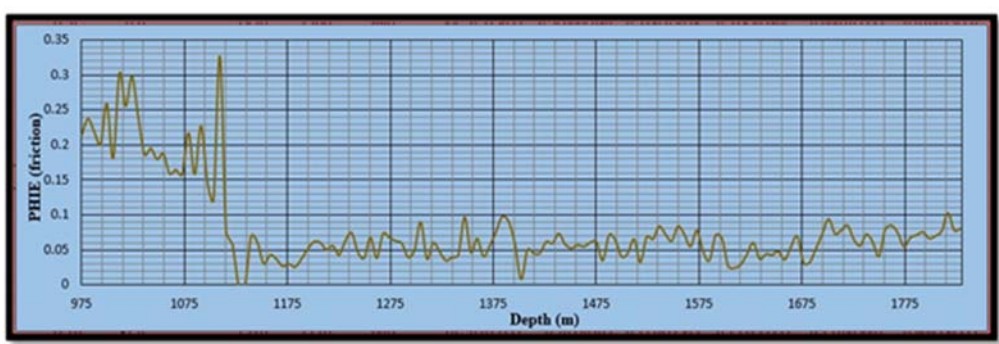

**Figure 20.** Effective porosity variation along the depth.

### 5.5. Saturation of Water

Saturation of water ($S_w$) is calculated by using Archie's equation [26]:

$$S_W = \sqrt{(R_W/(R_t * \varphi e^2))} \tag{5}$$

where $S_w$ is saturation of water, $R_w$ is resistivity of water, $R_t$ is resistivity of the true zone, and $\varphi_e$ is effective porosity; the log is shown in Figure 21.

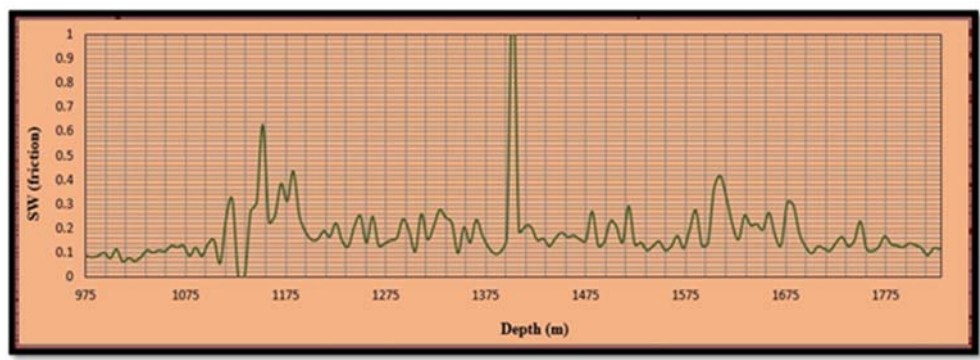

**Figure 21.** Water saturation along the depth.

### 5.6. Saturation of Hydrocarbon

Hydrocarbon saturation ($S_h$) is the most crucial feature due to the potential of the reservoir to generate hydrocarbons [27]. The result is shown in Figure 22, and the following formula was used for its calculation:

$$S_h = 1 - S_w. \tag{6}$$

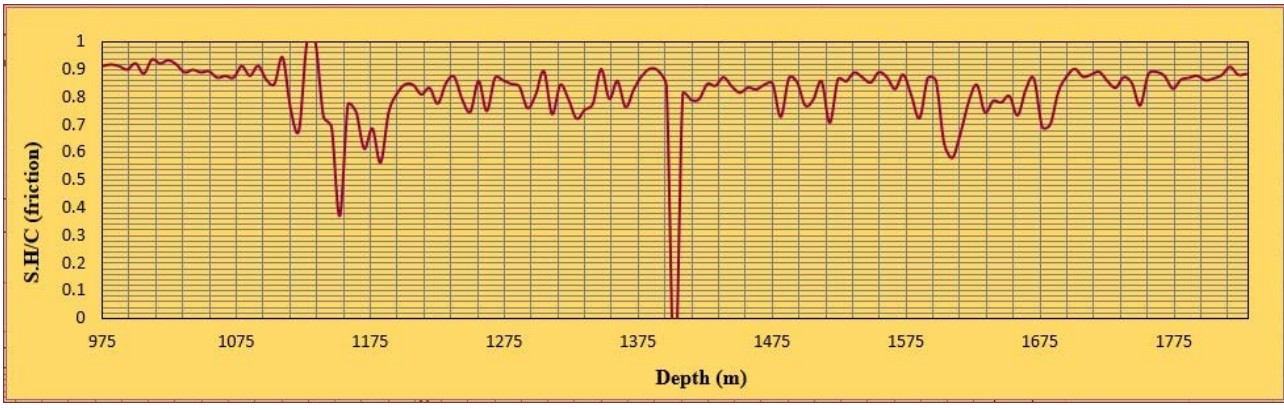

**Figure 22.** Hydrocarbon saturation along the depth.

In order to find out any hydrocarbon indicator (HI), all the well logs are shown together in Figure 23 in order to find out the hydrocarbon-bearing sand/carbonate and shale zones.

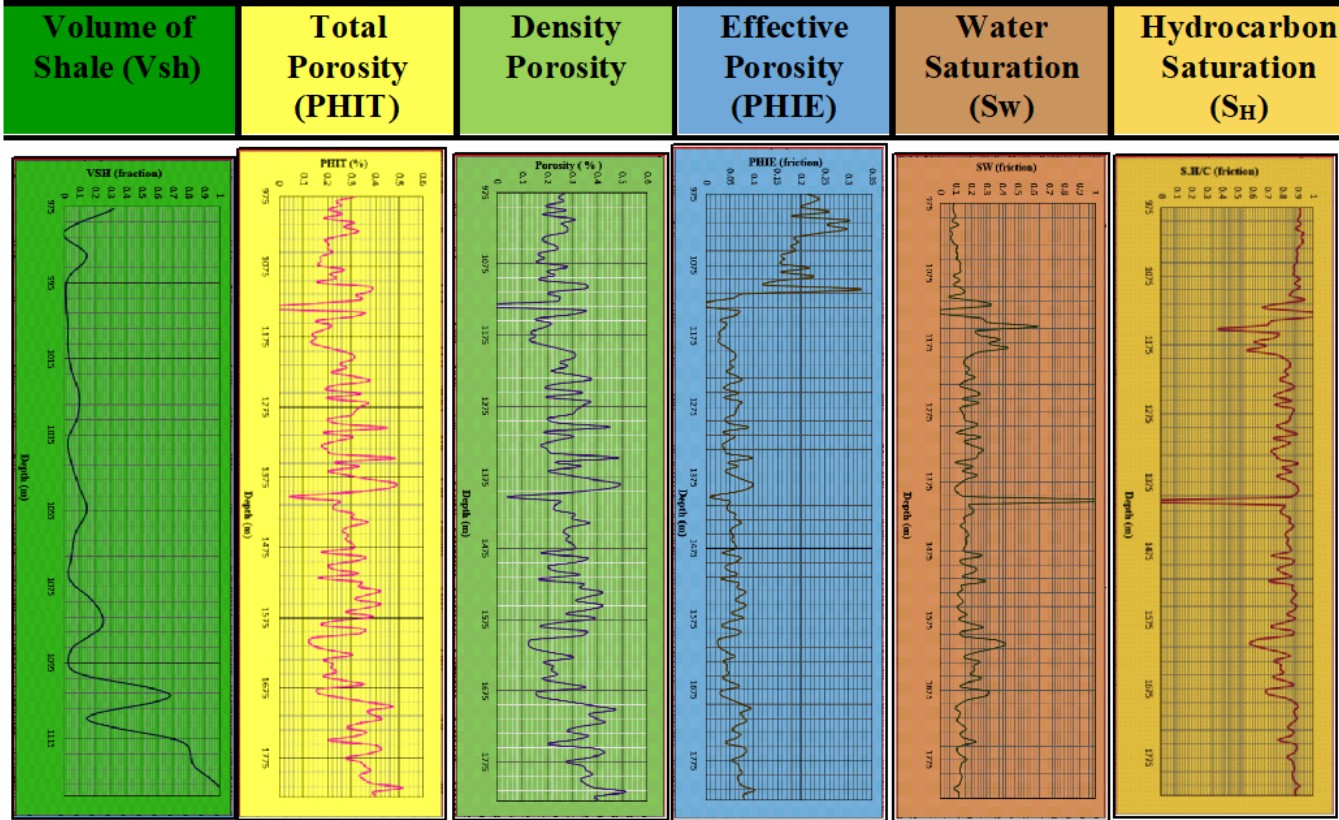

**Figure 23.** Combined well logs.

## 6. Discussion

In Figure 16, the volume of shale mostly varies from 40 to 80% up to the depth of 1115 m. This percentage shows that the lithology is partially clean and partially dirty and presents some presence of sand; therefore, it indicates a reservoir zone.

In Figure 20, effective porosity varies from 3 to 15% up to depths of 975 to 1115 m then decreases up to 1175 m, which shows that after 1115 m, compaction starts, which damages the interconnectivity of pores.

In Figure 21, the saturation of water lies almost in the range of 0–10%. Water saturation continuously changes with depth in this variation due to the fracture or compaction present in the formations. This is represented by the zone of less water saturation and high hydrocarbon saturation. Comparing Figures 21 and 22, hydrocarbon saturation and water saturation, we can clearly identify that these two graphs are inverse images of each other. This indicates that with increased hydrocarbon saturation, the water saturation decreases, and vice versa. This shows the zones of less water saturation and high hydrocarbon saturation.

## 7. Conclusions

The objective of the current research work was to interpret the subsurface structure and reservoir characterization of Kabirwala area Tola (01), located in Punjab platform, Central Indus Basin, utilizing 2D seismic and well log data. Formation evaluation for hydrocarbon potential using the reservoir properties was performed in this study. The outcomes of the current study are summarized as follows:

- The seismic data interpretation confirms that normal faults with horst and graben structures are favorable for the accumulation of hydrocarbons.
- The high zone present in the northwestern part of the contour maps could be a possible location of hydrocarbon entrapment, which is further confirmed by the presence of the Tola01 well.
- For the marked zone of interest, the average water saturation is 18%, average total porosity is 27.8%, average effective porosity is 8.2%, and net pay thickness is 84.14 m.
- The time section confirms that the formations are shallower in the northwestern direction and deeper along the southeastern direction. The stratigraphic conditions of these Jurassic age formations led to a complete petroleum system in the area.
- As a limitation, the current study needs further and suitable calibration with more actual well data to prove reliability and efficiency.
- In the future, we will implement 3D seismic data interpretation in the proposed field of our research. Similarly, we will also attempt to use the inversion algorithm for further exploration to obtain rich information.

**Author Contributions:** Conceptualization, N.A., S.K. and A.A.-S.; methodology, N.A., S.K. and A.A.-S.; software, N.A.; validation, N.A., S.K. and A.A.-S.; formal analysis, N.A. and S.K.; investigation, N.A., S.K. and A.A.-S.; resources, S.K. and A.A.-S.; data curation, N.A.; writing—original draft preparation, N.A., S.K. and A.A.-S.; visualization, N.A. and S.K.; supervision, S.K. and A.A.-S.; project administration, S.K. and A.A.-S.; funding acquisition, A.A.-S. All authors have read and agreed to the published version of the manuscript.

**Funding:** This research received funding from the College of Petroleum Engineering and Geosciences of KFUPM.

**Institutional Review Board Statement:** Not applicable.

**Informed Consent Statement:** Not applicable.

**Data Availability Statement:** The data are available by contacting the corresponding author.

**Acknowledgments:** The authors appreciate and acknowledge the support provided by King Fahd University of Petroleum and Minerals (KFUPM), KSA.

**Conflicts of Interest:** The authors declare no conflict of interest.

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
