# Peer review of "Seismic Data Interpretation and Petrophysical Analysis of Kabirwala Area Tola (01) Well, Central Indus Basin, Pakistan"

_applsci, doi:10.3390/app11072911_

Round 1

Reviewer 1 Report

Considering the previous revisions and the final manuscript, it needs improving on many aspects: text, figures, information and data.

The manuscript is incomplete in many parts; it contains statements that do not conform to the basic of geology e geophysics. There are figures that have no scientific value, others that are too dark and are not legible, and still others that describe the Tola well log that cannot be used for a correlation between the different parametric geophysics.

Consider that you:

  1. have made a seismic calibration using Tola Well but, where is Tola well?
  2. have talked about horst and graben using a time map;
  3. have used figures without units of measure and it is not clear what you are talking about;
  4. have used illegible dark figures like 8c, 9c 11c, 12c, 13c, 14c and 15c;
  5. haven't used a typical Geophysical Borehole Logging of Tola 01 Well.

According these considerations and if you want to modify this manuscript for publication in Applied Sciences, you must modify it and integrate it in the following points and include:

  • the location of the Tola Well;
  • a structural map of the study area at adequate scale;
  • the unit scale in each figure. It is not clear if we are talking about meters. seconds, milliseconds and so on;
  • the geophysical borehole logging of Tola -01 Well and please, cancel the figures from 17 to 32.

Author Response

The following represent point-by-point answers to the reviewers’ comments. Appropriate revisions are made in the revised manuscript, as explained hereunder. In the revised version of the manuscript, all the revisions are highlighted in yellow.

Comments by Reviewer 1:

  • Consider that you:
  1. have made a seismic calibration using Tola Well but, where is Tola well?
  2. have talked about horst and graben using a time map;
  3. have used figures without units of measure and it is not clear what you are talking about;
  4. have used illegible dark figures like 8c, 9c 11c, 12c, 13c, 14c and 15c;
  5. haven't used a typical Geophysical Borehole Logging of Tola 01 Well.

According these considerations and if you want to modify this manuscript for publication in Applied Sciences, you must modify it and integrate it in the following points and include:

  • the location of the Tola Well;
  • a structural map of the study area at adequate scale;
  • the unit scale in each figure. It is not clear if we are talking about meters. seconds, milliseconds and so on;
  • the geophysical borehole logging of Tola -01 Well and please, cancel the figures from 17 to 32.

Answer: The authors are thankful to the reviewer for the valuable comments. As suggested, in the revised version of the manuscript, all the above-mentioned comments and the comments given in the PDF document, are properly addressed. The following represent, point-by-point answers to the reviewers’ comments.

  • the location of the Tola Well;

The study area (Tola (01) Well), is located in the central Indus basin (CIB). The central Indus basin (CIB) consists of Punjab Platform, Suleiman Depression (Zindapir Inner Fold Zone, Mari Bugti Inner Fold Zone), along with Suleiman Fold Belt. The location of the study area is shown in Figures 1a, 1b and 2a. As shown in Figure 1a, Tola (01) well is situated in the Kabirwala area near district Multan. The Kabirwala extent is positioned in the Punjab Territory, the geographical relates are 30020’10” North and 70043’30” East. Kabirwala is one of the four tehsils in Khanewal district Multan. It lies in Central Indus Basin Punjab Platform, Pakistan.

Figure 1a. Location of the study area.

Figure 1b. Tectonic map of Central Indus Basin and subdivisions of petroleum zones.

Figure 2a point out the “Base Map” of the study area (Tola-01 well), located in the Punjab platform of central Indus Basin (CIB). The key purpose of the “Base Map” is to point out the particular direction of the seismic lines, shot points as well as the well position on the seismic line. In the current study, the “Base Map” point out two strike lines and three dip lines. The two strike lines have the shooting trend from north to south direction. However, the three dip lines have the shooting trend from east to west direction. Location of Tola (1) is added in the base map given below. The Tola (1) well is located at 375-KBR-230 line in the base map.

Figure 2a. Base map of the study area

  • a structural map of the study area at adequate scale;

The study area is located in the Kabirwala area near district Multan. It is positioned in central Indus basin. As shown in Figure 1b, the central Indus basin (CIB) and Upper Indus basin (UIB) are kept apart from each other in the North by Sargodha high and Pezu uplift. It is tighten from East through Indian Shield, from West by marginal zone of Indian plate and from South through Sukkur rift as shown in Figure 1b.

Figure 1b. Central Indus Basin and subdivisions into Petroleum zones (Raza et al, 1989).                         

  • the unit scale in each figure. It is not clear if we are talking about meters. seconds, milliseconds and so on;

As suggested, in the revised version of the manuscript, in all the figures, units are clearly mentioned. The suggested corrections have been made in Figures 8a, 8b, 9a, 9b, 10a, 10b, 11a, 11b, 12a, 12b, 13a, 13b, 14a, 14b, 15a and 15b. For time, seconds is used and for depth, meters is used.

  • the geophysical borehole logging of Tola -01 Well and please, cancel the figures from 17 to 32.

As suggested, in the revised version of the manuscript, the mentioned figures have been removed.

Finally, the authors wish to thank the reviewer for his constructive remarks, which are well-taken and implemented to improve the clarity and quality of the manuscript.

Reviewer 2 Report

Authors have tried to interpret the subsurface structures using the seismic data set. 

Main concerns are:

  1. It is unclear what is the source of the TVD chart. Converting the domain of seismic data set needs special consideration. If we use the TVD chart estimated from the sonic log, the calibration is essential. No information?
  2. In many Figures (4 to 7) some horizons are picked as known formations in the study area but it is unclear, are they top or base of FMs? To transfer top of formations/markers at the well location to seismic data set, it is unavoidable to carry out seismic-well tie which needs the source wavelet estimation and synthetic trace creation at the well location. Then try to find the best correlation between synthetic and real seismic. No evidence of domain change or correlation has been found in the paper.
  3. In Figures 5 to 7, many faults are picked. No-fault displacements are visible at the picked locations.  
  4. I could not find the location of drilled well in the time structural map.
  5. To find out any hydrocarbon indicator (HI) it is essential to make cross plots or show all the well logs together to find out the hydrocarbon-bearing sand/carbonate and shale zones. 
  6. The definition of all well logs can be found in the paper but no good interpretation has been done. 
  7. Based on the caliper it is possible to find out the washout zone and the effect of that on gamma-ray. 
  8. It is so important to show the horizon slice at the target and find out the amplitude/phase changing at the top of the structure that can be an indicator of hydrocarbon or lithology changing. 
  9. No stratigraphic interpretation has been done. 
  10. High fluctuation at the Dunghan horizon slice can be due to automatic picking and no quality control has been done to correct the issues (Figs 8b and 8c). 
  11. The same issue repeated for other horizon slices (Figs 10, 12, 13).

Author Response

The following represent point-by-point answers to the reviewers’ comments. Appropriate revisions are made in the revised manuscript, as explained hereunder. In the revised version of the manuscript, all the revisions are highlighted in green.

Comments by Reviewer 2:

  • It is unclear what is the source of the TVD chart. Converting the domain of seismic data set needs special consideration. If we use the TVD chart estimated from the sonic log, the calibration is essential. No information?

Answer: The authors are thankful to the reviewer for the valuable comment. As suggested, in the revised version of the manuscript, the details has been added related to the TVD chart on pages 12 & 13 of the revised manuscript and is highlighted in green.

  • In many Figures (4 to 7) some horizons are picked as known formations in the study area but it is unclear, are they top or base of FMs? To transfer top of formations/markers at the well location to seismic data set, it is unavoidable to carry out seismic-well tie which needs the source wavelet estimation and synthetic trace creation at the well location. Then try to find the best correlation between synthetic and real seismic. No evidence of domain change or correlation has been found in the paper.

Answer: The authors are thankful to the reviewer for the valuable comment. As suggested, in the revised version of the manuscript, the domain change and the correlation are discussed in detail on pages 11 & 12 of the revised manuscript and is highlighted in green.

  • In Figures 5 to 7, many faults are picked. No-fault displacements are visible at the picked locations. 

Answer: The authors are thankful to the reviewer for the valuable comment. In the revised version of the manuscript, it is clear from the mentioned figures that there is significant displacement of the reflectors across both sides the faults.

  • I could not find the location of drilled well in the time structural map.

Answer: The authors are thankful to the reviewer for the valuable comment. The well location is shown in the base map, given below that can be correlated with the time structure map. The Tola (1) well is located at 875-KBR-230 line in the base map.

  • To find out any hydrocarbon indicator (HI) it is essential to make cross plots or show all the well logs together to find out the hydrocarbon-bearing sand/carbonate and shale zones. 

Answer: The authors are thankful to the reviewer for the valuable comment. In order to find out any hydrocarbon indicator (HI), all the well logs are shown together in Figure 23, in order to find out the hydrocarbon-bearing sand/carbonate and shale zones and is discussed in the discussion section of the revised manuscript.

  • The definition of all well logs can be found in the paper but no good interpretation has been done. 

Answer: The authors are thankful to the reviewer for the valuable comment. Based on the comments from the “Reviewer 1”, the section containing the definition of all well logs and the supporting figures have been removed from the revised manuscript.

  • Based on the caliper it is possible to find out the washout zone and the effect of that on gamma-ray.

Answer: The authors are thankful to the reviewer for the valuable comment. Based on the comments from the “Reviewer 1”, the section related to the caliper, the gamma-ray and the supporting figures have been removed from the revised manuscript.

  • It is so important to show the horizon slice at the target and find out the amplitude/phase changing at the top of the structure that can be an indicator of hydrocarbon or lithology changing.

Answer: The authors are thankful to the reviewer for the valuable comment. The time contour map of the Dunghan formation shows a monocline structure dipping towards the east direction. The line also shows that there is minor faulting in this formation, therefore, it implies that the Dunghan formation is gently dipping as a monocline structure of the Punjab platform. The time increases from 0.89 to 1.17 seconds and is shown in figure 8b.

The time contour map of Samana Suk formation also pointed towards the monocline structure dipping moderately east-ward direction. The increase in reflection time is from 0.97 to 1.19 sec. The time contour surface of Samana Suk formation were generated by joining the points of equal time. These time values have assign a color bar which can be used as a guide for interpretation. On the color bar, the red color shows the shallowest point while the purple blue color shows the deepest point.

The time contour map of Datta formation also shows the monocline structure like the other two formations, descripting the characteristics of the Punjab Platform. The increase in reflection time is from 1.02 to 1.28 sec.

The time contour map of Warcha Sand stone also pointed towards the monocline structure dipping moderately to the eastward direction. The increase in reflection time is from 1.34 to 1.64 sec. The depth contour map of the Warcha formation gives the same scenario about the dipping monocline structure of the Punjab platform area of the central Indus Basin.

  • No stratigraphic interpretation has been done.

Answer: The authors are thankful to the reviewer for the valuable comment. Yes, in the current study, the stratigraphic interpretation has not performed because it was out of the scope of the current study. The stratigraphic interpretation will be performed as a future extension of the current study.

  • High fluctuation at the Dunghan horizon slice can be due to automatic picking and no quality control has been done to correct the issues (Figs 8b and 8c).

Answer: The authors are thankful to the reviewer for the valuable comment. The time contour map of the Dunghan formation shows a monocline structure dipping towards the east direction. The line also shows that there is minor faulting in this formation, therefore, it implies that the Dunghan formation is gently dipping as a monocline structure of the Punjab platform. The time increase from 0.89 to 1.17 seconds as shown in figure 8b. The 3D view of the particular formation surface helps in comprehending the monocline structure in the Punjab platform region northward.

Figure 8b. Time surface map of Dunghan formation.

  • The same issue repeated for other horizon slices (Figs 10, 12, 13).

Answer: The authors are thankful to the reviewer for the valuable comment. The time contour map of the Datta formation gives the picture of monocline structure as by the other two formations, depicting the features of the Punjab platform. The reflection time increases from 1.06 seconds to 1.4 seconds that is shown in the figure 10b.

Figure 10b. Time surface map of Datta formation.

The depth contour map of the Dunghan formation gives the same scenario about the dipping monocline structure of the Punjab platform area of the central Indus Basin. The depth value of the Dunghan formation is dipping towards eastwards direction and the depth values are decreasing from 1295 to 920 meter that is shown in the figure 12b.

Figure 12b. Depth surface map of Dunghan formation.

The time contour map of the Samana Suk formation also implies towards the monocline structure dipping gently towards the east direction. The reflection time increase from 0.01 to 1.28 sec that is show in the figure 13b.          

Figure 13b. Depth surface map of Samana Suk formation.

Finally, the authors wish to thank the reviewer for his constructive remarks, which are well-taken and implemented to improve the clarity and quality of the manuscript.

Round 2

Reviewer 1 Report

Dear authors,

according to your last revision the manuscript was improved and a lot of data was clarified and the integrated information helped understanding the Seismic Data Interpretation and Petrophysical Analysis of the Kabirwala Area.  

This manuscript is a resubmission of an earlier submission. The following is a list of the peer review reports and author responses from that submission.

Round 1

Reviewer 1 Report

Specifically:

1/ The results of the seismic and the boreholes are badly presented, the seismic faring slightly better than the boreholes. The latter are simply dumped in the paper; the interpretation is just one line in the conclusions. It would be far better to have a complete interpretation chapter.

2/ The diagrams/figures are badly constructed. Not all figures are required, for instance a 3D surfer Image next to each depth map is unjustified. Furthermore, the method used to construct the depth maps is not given, but I assume some krieging algorithm. I cannot see how the time contour maps could have been made from the seismic. There are no scale units on the maps but I assume ms TWT.  The seismic profiles seem to be screenshots, but the depth scale is unreadable and there are no units, I cannot read the horizontal scale.

3/ the borehole data is disgracefully mistreated. Each borehole log (where is the Tola-01 well?) is first treated like a textbook entry (I hope this wasn't copied), which is completely unnecessary. Then we see a plot of the property, but without any Interpretation.

4/ the abstract does not inform the reader about the specific aim of the study.

5/ If you use Corel draw, like you say in the Appendix, why is Figure 1 so badly constructed? Trace the important information, leaving out the unnecssary data. It is unexcusable to just dump Google Map pieces together like this.

6/ the English is poor to the extent that I cannot understand certain paragraphs.

7/ Much more could be said about the seismic interpretation. For instance, the thickness changes of some of the beds (and not in others) in the grabens says alot about the timing of the rifts.

Author Response

The following represent point-by-point answers to the reviewers’ comments. Appropriate revisions are made in the revised manuscript, as explained hereunder.

 Comments by Reviewer 1:

  • The results of the seismic and the boreholes are badly presented, the seismic faring slightly better than the boreholes. The latter are simply dumped in the paper; the interpretation is just one line in the conclusions. It would be far better to have a complete interpretation chapter.

Answer: As suggested, in the revised version of the manuscript, a complete interpretation has been presented from line 278 to 288 and line 298 to 302 on page 9 and is highlighted in yellow. Figure 1d is also added for further explanation and is highlighted in yellow.

  • The diagrams/figures are badly constructed. Not all figures are required, for instance a 3D surfer Image next to each depth map is unjustified. Furthermore, the method used to construct the depth maps is not given, but I assume some krieging algorithm. I cannot see how the time contour maps could have been made from the seismic. There are no scale units on the maps but I assume ms TWT. The seismic profiles seem to be screenshots, but the depth scale is unreadable and there are no units, I cannot read the horizontal scale.

Answer: As suggested, in the revised version of the manuscript, the two-way time is mentioned in Table 2 and depth time is mentioned in Tables 4 & 5 is highlighted in yellow. All the figures have been revised and the quality has been improved. Appropriate scaling has been included in each figure.

  • The borehole data is disgracefully mistreated. Each borehole log (where is the Tola-01 well?) is first treated like a textbook entry (I hope this wasn't copied), which is completely unnecessary. Then we see a plot of the property, but without any Interpretation.

Answer: As suggested, in the revised version of the manuscript, a complete interpretation has been presented in section 4 on page 9 and is explained using figure 1d and is highlighted in yellow.  Appropriate scaling has been included in figures 3 to 7 and is highlighted in yellow.

  • The abstract does not inform the reader about the specific aim of the study.

Answer: In the revised version of the manuscript, the abstract has been revised to increase the understanding of the specific aim of the study and is highlighted in yellow.

  • If you use Corel draw, like you say in the Appendix, why is Figure 1 so badly constructed? Trace the important information, leaving out the unnecssary data. It is unexcusable to just dump Google Map pieces together like this.

Answer: As suggested, in the revised version of the manuscript, Figure 1a has been revised and an effort has been made to trace the important information and is highlighted in yellow.

  • The English is poor to the extent that I cannot understand certain paragraphs.

Answer: In the revised version of the manuscript, the English has been excessively improved. Various paragraph has been rephrased to increase the readability of the manuscript.

  • Much more could be said about the seismic interpretation. For instance, the thickness changes of some of the beds (and not in others) in the grabens says alot about the timing of the rifts.

Answer: As suggested, in the revised version of the manuscript, a complete interpretation has been presented from line 278 to 288 and line 298 to 302 on page 9 and is highlighted in yellow. The strata in an exposure or outcropping of sedimentary rock can range from layers as thin as paper, known as lamina (plural: laminae or laminations) to beds tens of feet thick. Generally, the more stable and consistent the environmental conditions during deposition, the thicker will be the strata.

Finally, the authors wish to thank the reviewer for his constructive remarks, which are well-taken and implemented to improve the clarity and quality of the manuscript.

Reviewer 2 Report

Your manuscript is very interesting in order to understand the structural setting of the area and to increase the knowledge about the geometry of the reservoir.

According to this and considering the value of your research, it would be advised to review and to improve the manuscript and to give more details of your study.

Generally speaking, you need to enrich the Introduction section because there’s a good amount of information about It in the geological references. You must mention the main authors that have described the stratigraphy, the structural setting and the petroleum system of the area.

Finally, by using time maps it is not possible to evaluate the hydrocarbon flow and the geometry of the reservoir; to do this, you have to use the maps in depth.

Comments

Please include:

  • a structural map of the study area;
  • the location of the Tola Well;
  • The Stratigraphy of the area using figure and text;
  • Verify the interpretation of the extensional faults;
  • Reconsider Line 193 and so on, because this statement is almost impossible using time map: On this time surface, the probable hydrocarbon flow direction was marked, which shows the location of the hydrocarbon accumulation as shown in Figure 8c;
  • Use the formal name: Samana Suk Formation and not Samana Formation;
  • Use the formal name: Warcha Formation or Warcha Sandstone;
  • Include the unit scale in each figure;
  • In depth maps of the hydrocarbon flow direction;
  • A table with the petrophysical parameters of the Formations considered;
  • A new figure, where it is possible, to see the correlation from stratigraphy and the petrophysical parameters.

Author Response

The following represent point-by-point answers to the reviewers’ comments. Appropriate revisions are made in the revised manuscript, as explained hereunder.

Comments by Reviewer 2:

  • A structural map of the study area

Answer: As suggested, in the revised version of the manuscript, a structural map has been added as shown in figure 1b and is highlighted in bright green.

  • The location of the tola well;

Answer: As suggested, in the revised version of the manuscript, the location of the tola well has been shown in figures 1a & 1b.

  • The stratigraphy of the area using figure and text

Answer: As suggested, in the revised version of the manuscript, a complete stratigraphy has been presented from line 101 to line 271 on pages 5 to 8 and is highlighted in bright green. Figures 1c is also added for further explanation.

  • Verify the interpretation of the extensional faults;

Answer: As suggested, in the revised version of the manuscript, the interpretation of the extensional faults has been added from line 288 to line 294 on page 9 and is highlighted in bright green color.

  • Reconsider line 193 and so on, because this statement is almost impossible using time map: On this time surface, the probable hydrocarbon flow direction was marked, which shows the location of the hydrocarbon accumulation as shown in Figure 8c;

Answer: As suggested, in the revised version of the manuscript, the statement has been revised as given from line 404 to line 413 on page 15 and is highlighted in bright green color.

  • Use the formal name: Samana Suk Formation and not Samana Suk Formation;

Answer: As suggested, in the revised version of the manuscript, the name has been corrected from “Samana Suk Formation” to “Samana Suk Formation”.

  • Use the formal name: Warcha Formation or Warcha Sandstone;

Answer: As suggested, in the revised version of the manuscript, the formal name “Warcha Formation” has been used throughout the manuscript.

  • Include the unit scale in each figure;

Answer: As suggested, in the revised version of the manuscript, the unit scale in all the figures has been updated.

  • In depth maps of the hydrocarbon flow direction;

Answer: In the revised version of the manuscript, the in-depth maps of the hydrocarbon flow direction are given in sections 4.8, 4.9, 4.10 and 4.11.

  • A table with the petrophysical parameters of the formations considered;

Answer: As suggested, in the revised version of the manuscript, the petrophysical parameters has been presented in table 7 and is highlighted in bright green.

  • A new figure, where it is possible, to see the correlation from stratigraphy and the petrophysical parameters.

Answer: As suggested, in the revised version of the manuscript, figures 16 to 22 and table 7 has been added to present and summarize the various petrophysical parameters and their variations along the depth.

Finally, the authors wish to thank the reviewer for his constructive remarks, which are well-taken and implemented to improve the clarity and quality of the manuscript.

Reviewer 3 Report

This paper presents a worth topic investigation.  The author has pretty good work on an research approach. But there are some issues have to be addressed before accepting:

  • The literature review should be organized in a better format; I would suggest you use table summarizes findings from the literature review
  • Some implication or data support from theory and field test is highly recommended, but not necessary, but at least discuss it in the paper.
  • Limitation of this research needed to be addressed and discussed
  • some figure's quality need improve, such as figure 7
  • The future plan or suggestion is highly recommended.
  • More related literature should be cited and discussed. The list of paper is shown below.
  • A Study on multiple time-lapse seismic AVO inversion, Chinese Journal of Geophysics , 2005,48(4), pp. 902-908
  • Lithofacies distribution and gas-controlling characteristics of the wufeng-longmaxi black shales in the southeastern region of the Sichuan Basin, China, Journal of Petroleum Science and Engineering, 2018 165,269-283
  • Layout optimization of large-scale oil-gas gathering system based on combined optimization strategy. Neurocomputing, 332:159-183.
  •  

Author Response

Comments by Reviewer 3:

  • The literature review should be organized in a better format; I would suggest you use table summarizes findings from the literature review

Answer: As suggested, in the revised version of the manuscript, the table 1, summarizing the findings of the literature review has been added in section 1 on pages 2 & 3 of the manuscript and is highlighted in turquoise color.

  • Some implication or data support from theory and field test is highly recommended, but not necessary, but at least discuss it in the paper.

Answer: As suggested, in the revised version of the manuscript, the input data has been discussed in section 1 on page 3 of the manuscript is highlighted in turquoise color.

  • Limitation of this research needed to be addressed and discussed

Answer: As suggested, in the revised version of the manuscript, the limitation of the current research has been discussed in section 6 on page 30 of the manuscript and is highlighted in turquoise color.

  • Some figures quality need improve, such as figure 7

Answer: In the revised version of the manuscript, the quality of all the figures has been improved.

  • The future plan or suggestion is highly recommended.

Answer: In the revised version of the manuscript, the future plan has been discussed in section 6 on page 30 and is highlighted in turquoise color.

  • More related literature should be cited and discussed.

Answer: In the revised version of the manuscript, the suggested papers has been cited as; [1], [2] and [4] and are highlighted in turquoise color.

Finally, the authors wish to thank the reviewer for his constructive remarks, which are well-taken and implemented to improve the clarity and quality of the manuscript.

Round 2

Reviewer 1 Report

While is true that some improvements have been made to the manuscript (stratigraphy especially), the authors have still not answered the points I made in my last review. This is also apparent in the cover letter.

Specifically,

1/ The base map of the seismic lines shows that it is problematic to create a 3D map because coverage is not homogeneous. In particular, on the seismic lines data density is too high, and too low in the SE corner. This is not taken in to account when creating the contour maps. Consequently I don't think the contour maps, (and therefore  the surface maps or flow maps) make any sense. It would make more sense to define the faults as breaklines in the surfer file. Please give the algorithm use to grid the data.

3/ The English still needs to be improved. Not only is the grammar and spelling bad, but the style of English is condescending
For instance:
"The purpose of the interpretation is to understand how these structures are formed and is necessary for mapping and marking the horizons above and below the target zone. The strata in an exposure or outcropping of sedimentary rock can range from layers as thin as paper, known as lamina (plural: laminae or laminations) to beds tens of feet thick. Generally, the more stable and consistent the environmental conditions during deposition, the thicker will be the strata."
This sound like it taken from a text book.

4/ In seismic line 875-KBR-221, all the beds are 0.3 s TWT deeper. Why? What about tie points at the intersection of the seismic lines - don't they tell you whether the interpretation is correct over the whole area?

5/ Table 2: what is strike line and dip line meant in this context?

6/ after the borehole geophysics, there must be discussion of the results of all the data. This is missing completely.

Reviewer 2 Report

Considering your work, I compared the two revisions and I also read the comments of the other two reviewers.

Using all this information, below you can find more comments regarding the manuscript:

  • Forgot to consider my comments made in the attached file (pdf);
  • Forgot to improve the quality of some figures;
  • Didn’t include the location of the Tola well in the base map (e.g. fig. 1a, 1b and 2a);
  • Didn’t include the scale in the figures from 8a to 15c;
  • Didn’t modify the numbers of the figures 12a and 12b (you wrote 13a and 13b);
  • Didn’t modify the symbol of the fault of the figures from 3 to 7. The fault is a line;
  • Didn’t improve the correlation from the well and the seismic line and I don’t understand if they are talking about the same area;
  • Didn’t include a borehole log of the Tola well (see attached file);
  • Didn’t describe the interpretation of the seismic line in time and in depth and how the stratigraphy is changing in the basin and how this can drive the oil migration.

I encourage you to rewrite your manuscript using the all the comments of the reviewers because your work is very interesting to present to our community, but it needs improvements.

Applied Sciences is a very important journal with a high Impact Factor, and I encourage to submit your work as soon as possible.
